# G9a/GLP inhibition during ex vivo lymphocyte expansion increases in vivo cytotoxicity of engineered T cells against hepatocellular carcinoma

Maxine S. Y. Lam[1], Jose Antonio Reales-Calderon[1], Jin Rong Ow[1], Joey J. Y. Aw[1], Damien Tan[1], Ragavi Vijayakumar[1], Erica Ceccarello[1], Tommaso Tabaglio[1], Yan Ting Lim[2], Wang Loo Chien[2], Fritz Lai[1], Anthony Tan Tanoto[3], Qingfeng Chen[1], Radoslaw M. Sobota[2,4], Giulia Adriani[5,6], Antonio Bertoletti[3], Ernesto Guccione[7] & Andrea Pavesi[1,8] ✉

Engineered T cells transiently expressing tumor-targeting receptors are an attractive form of engineered T cell therapy as they carry no risk of insertional mutagenesis or long-term adverse side-effects. However, multiple rounds of treatment are often required, increasing patient discomfort and cost. To mitigate this, we sought to improve the antitumor activity of transient engineered T cells by screening a panel of small molecules targeting epigenetic regulators for their effect on T cell cytotoxicity. Using a model for engineered T cells targetting hepatocellular carcinoma, we find that short-term inhibition of G9a/GLP increases T cell antitumor activity in in vitro models and an orthotopic mouse model. G9a/GLP inhibition increases granzyme expression without terminal T cell differentiation or exhaustion and results in specific changes in expression of genes and proteins involved in pro-inflammatory pathways, T cell activation and cytotoxicity.

Adoptive cell transfer (ACT) has been highly effective in targeting certain refractory cancers and remains potentially effective for other cancers as new targets of cancer-immune interactions are identified[1]. ACT involves isolating immunocompetent cells from cancer patients, expanding them ex vivo, and infusing them back into the patient. Cells used for ACT can be unmodified tumor-infiltrating lymphocytes (TILs) isolated from the patient or effector cells, typically T cells, isolated from the patient's peripheral blood then engineered to target the

tumor by incorporating a T cell receptor (TCR) or chimeric antigen receptor (CAR), with additional modifications to improve immune cell proliferation and persistence[2].

While cell therapies for various lymphomas have led to dramatic tumor regressions and have FDA approval, response in solid tumors remains varied. Solid tumors present unique challenges, such as tumor heterogeneity and the lack of tumor-specific targets, but also an immunosuppressive tumor microenvironment (TME) characterized by

[1]Institute of Molecular and Cell Biology, Agency for Science, Technology and Research (A*STAR), 138673 Singapore, Singapore. [2]Functional Proteomics Laboratory, SingMass National Laboratory, Institute of Molecular and Cell Biology, Agency for Science, Technology and Research (A∗STAR), 138673 Singapore, Singapore. [3]Duke-NUS Medical School, 169857 Singapore, Singapore. [4]Bioinformatics Institute, Agency for Science, Technology and Research (A∗STAR), 138671 Singapore, Singapore. [5]Singapore Immunology Network, Agency for Science and Technology (A*STAR), 138648 Singapore, Singapore. [6]Department of Biomedical Engineering, National University of Singapore, 117583 Singapore, Singapore. [7]Department of Oncological Sciences and Pharmacological Sciences, Center for Therapeutics Discovery, Tisch Cancer Institute, Icahn School of Medicine at Mount Sinai, 10029 New York, USA. [8]Mechanobiology Institute, National University of Singapore, 117411 Singapore, Singapore. ✉e-mail: andreap@imcb.a-star.edu.sg

poor T cell infiltration and terminal T cell differentiation and exhaustion at the tumor[3]. To overcome the hostile TME, lymphodepleting regimens combined with infusion of large numbers of the T cells is often used[3]. However, introducing large numbers of T cells significantly increases the risk of on-target/off-target toxicity, neurotoxicity, and cytokine release syndrome[1,4]. The use of engineered T cells that transiently express TCRs or CARs by mRNA gene transfer, reduces these risks. Additionally, they avoid the use of viral vectors, hence there is no risk of insertional mutagenesis and can be manufactured more easily at low cost and on a larger scale[5]. Such transient engineered T cells have shown significant antitumor activity in a phase I clinical trial[6] and in preclinical animal models[7,8]. However, multiple infusions are often required, increasing patient discomfort and treatment cost. To minimize this, strategies to improve the antitumor activity of transient engineered T cells are required.

Using small molecule inhibitors is a simple and cost-effective method for modulation of T cell behavior and cell therapy efficacy[9], and has been shown to limit terminal differentiation and promote proliferation when used during engineered T cell production[10–14]. T cell subpopulations are transcriptionally regulated, and such rapid and precise differentiation is thought to be epigenetically controlled[15,16]. Epigenetic inhibitors are a unique class of small molecule inhibitors with the potential to induce heritable changes in the epigenome and therefore persistent change in T cell behavior[16]. In this study, we screen a panel of epigenetic inhibitors for their effect on T cell cytotoxicity when added during ex vivo T cell expansion, using a model for transient TCR-T cells targeting hepatocellular carcinoma (HCC)[7]. We find that short-term G9a/GLP inhibition during the production of transiently engineered T cells improves antitumor cytotoxicity.

## Results

### G9a/GLP inhibitors increase the cytotoxicity of TCR-engineered T cells

Engineered transient-expressing TCR[+] T cells have previously been shown to be effective against hepatitis B virus-positive (HBV[+]) hepatocellular carcinoma[7]. We therefore used this model to screen for small molecule inhibitors targeting epigenetic regulators that could increase TCR[+] T cell cytotoxicity (Fig. 1A). 24 pre-selected small molecule inhibitors, were obtained from Structural Genomics Consortium (SGC) Open Chemistry Networks platform (Open Chem Networks). T cells derived from healthy donors were isolated and treated with drugs for 5 days, in accordance with existing clinical practices for expanding patient T cells for T cell therapy[7]. After which, drugs were withdrawn from T cells and T cells were transfected with HBV envelope S183–191-specific TCR mRNA by electroporation to generate TCR[+] T cells before use[7]. With this protocol we were able to achieve 50–90% TCR expression in T cells. TCR[+] T cell toxicity against the target cell line, HepG2 transduced to express the preS1 portion of the genotype D HBV envelope protein gene[17] (HepG2-preS1), was evaluated with 2D CellTox assays where fluorescence correlates to cell death. From the panel, A366 resulted in the highest increase in fluorescence (Fig. 1B). A366 is a selective inhibitor of G9a/GLP[18], a hetero or homodimeric enzyme mainly responsible for mono- and di-methylation of lysine 9 of histone H3 (H3K9me1/2)[19,20].

To confirm the effect of G9a/GLP inhibition on T cell viability and cytotoxicity, we selected two other chemically distinct compounds that selectively inhibit G9a/GLP, UNC0638[21] and UNC0642[22]. We identified a concentration that did not affect T cell proliferation (Supplementary Fig. 1A), and resulted in effective inhibition of G9a/GLP activity, as assessed by a reduction of H3K9me2 levels (Supplementary Fig. 1B, C). We observed a greater decrease in H3K9me2 levels in UNC0638-treated T cells, consistent with previous reports of its higher IC$_{50}$[21]. To better separate the effects on T cell cytotoxicity, TCR[+] T cell cytotoxicity

after treatment with the 3 different drugs was evaluated using a 3D microfluidic device (AIM Biotech, Singapore) according to previous studies[23] (Fig. 1C). This is because in a 3D cytotoxicity assay, T cells must actively migrate to target cells, while in 2D cytotoxicity assays effector-target cell interaction is largely mediated by gravity and leads to an overestimation of cytotoxicity[17,23]. Briefly, HepG2-preS1 cells were seeded in a collagen matrix in microfluidic channels, and TCR[+] T cells were introduced into one channel and allowed to migrate into the matrix toward target cells, and the proportion of target cells killed by T cells was quantified after 24 h. While TCR[+] T cell cytotoxicity generally improved after G9a/GLP inhibition, TCR[+] T cell cytotoxicity did not improve for some donors with UNC0638 and A366 (Fig. 1C). UNC0642-treated T cells showed the most consistent and greatest increase in target cell killing (Fig. 1C). UNC0642 also has a better pharmacokinetic profile than UNC0638 or A366[22], which might be relevant for cell therapy protocols that co-inject drugs with engineered T cells; hence, UNC0642 was used in further experiments. Previous work also suggests there is some cross-talk between G9a/GLP and Polycomb Repressive Complex 2 (PRC2), which is responsible for methylation of lysine 27 of histone H3 (H3K27)[24]. Consistent with other studies, we found no change in H3K27me3 after UNC0642 treatment (Supplementary Fig. 1D)[25–27].

To further verify the effect of UNC0642 treatment, we repeated the 2D (Fig. 1D–G) and 3D (Fig. 1H–J) cytotoxicity assays with additional controls and donors. TCR[+] T cell cytotoxicity was assayed in 2D in real time during co-culture with target cells using the xCELLigence RTCA DP (Fig. 1D–F), which uses impedance to measure target cell adhesion and growth. Continuous assay of cytotoxicity is particularly useful when using a transiently-expressed TCR[+] T cell in order to understand persistence of engineered T cell cytotoxicity. Untreated and UNC0642-treated non-transduced mock electroporated (TCR[-]) T cells were used as an additional control. Target cell growth was monitored for 48 h after T cell addition, since electroporated TCR[+] T cells lose ~60% of their TCR expression at 72 h post-electroporation (Supplementary Fig. 1E, F)[7] and TCR[+] T cells are added to target cells 24 h after transduction. UNC0642 treatment did not affect TCR expression (Supplementary Fig. 1G).

As early as 3.5 h after T cell addition, TCR[+] T cells restricted HepG2-preS1 cell growth more than TCR[-] T cells (Fig. 1E), indicating that transduction of TCR expression improved T cell targeting and cytotoxicity as previously reported[7]. The effect of TCR engineering became more pronounced over time, as target cell growth eventually recurred in TCR[-] T cell conditions while target cell growth continually decreased in TCR[+] T cell conditions (Fig. 1D, F, G). UNC0642-treated TCR[+] and TCR[-] T cells sustained an increase in cytotoxicity compared to untreated TCR[+] and TCR[-] T cells throughout the observed period (Fig. 1D–G), indicating that UNC0642 treatment increased T cell cytotoxicity regardless of engineered TCR expression status. The effect of UNC0642 treatment was more significant at lower effector: target ratios of engineered TCR[+] T cells (Supplementary Fig 1H–L), suggesting that UNC0642 was able to improve the efficiency of engineered T cells, especially at sub-optimal conditions.

Using the 3D cytotoxicity assay, we evaluated both the cytotoxicity and migration of TCR[+] T cells (Fig. 1H–J). Similar to our observations in 2D, TCR expression improved target cell killing in 3D, and UNC0642-treated TCR[+] T cells resulted in increased target cell killing (Fig. 1I). However, UNC0642 treatment did not increase cytotoxicity of mock electroporated TCR[-] T cells or naïve non-electroporated T cells (Fig. 1I). TCR expression improved T cell migration into the extracellular matrix (Fig. 1J), however there was no difference in T cell migration for UNC0642-treated TCR[+] T cells, suggesting that UNC0642 treatment increases the intrinsic cytotoxicity of T cells rather than their ability to migrate. It is possible that in TCR[-] T cells and

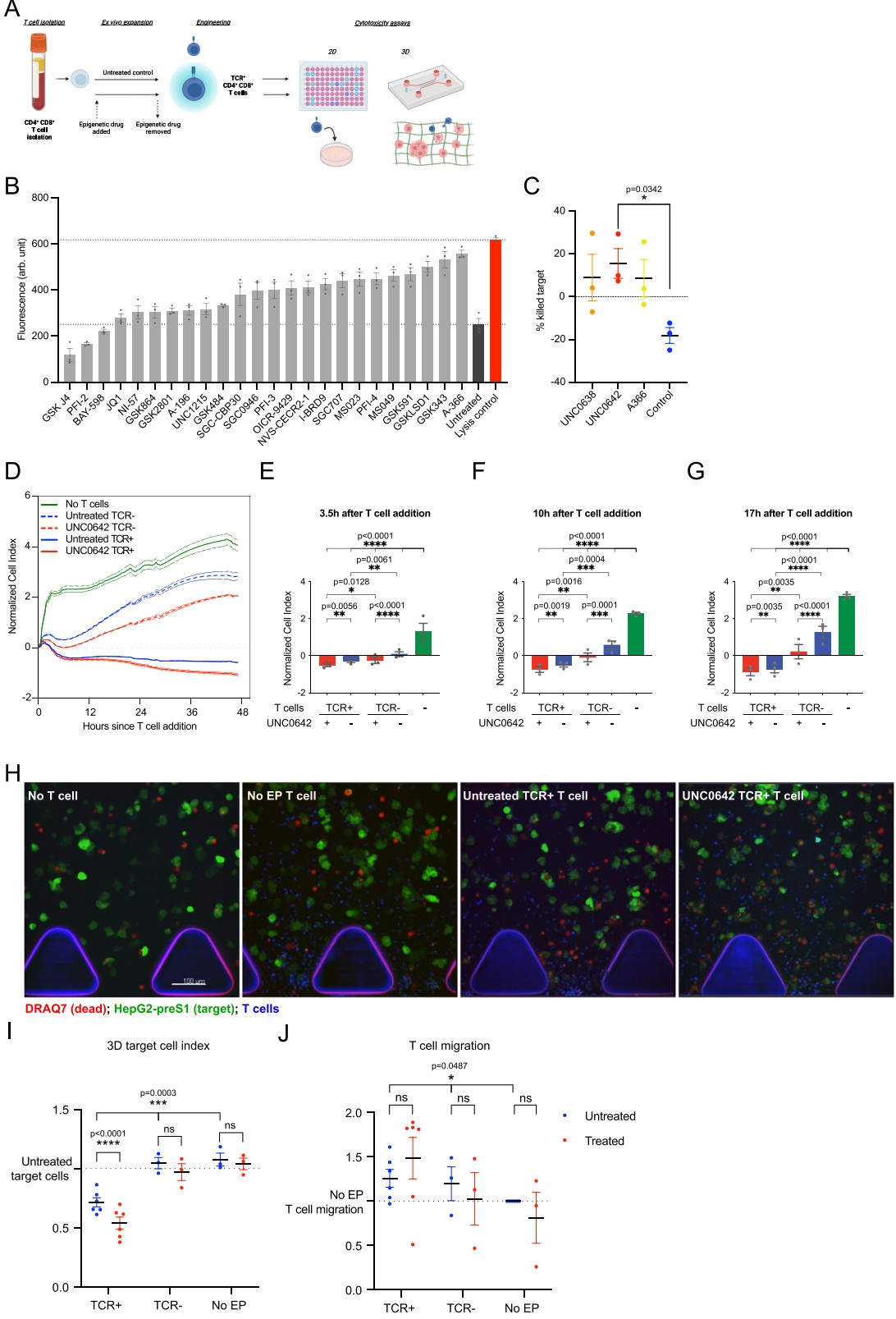

T cells without electroporation that despite being able to migrate into the extracellular matrix, that they are unable to target cells effectively without TCR expression, since there was no observed target cell death regardless of UNC0642 treatment (Fig. 1I).

Overall, the data indicate that G9a/GLP inhibition during T cell expansion increases TCR⁺ T cell target cell killing efficiency by increasing cytotoxicity.

## The G9a/GLP inhibitor UNC0642 increases the expression of markers of T cell activation and effector function without affecting T cell subpopulations

Epigenetic factors regulate T cell differentiation potential and phenotype[28], and T cell subpopulations such as CD8⁺ and cytotoxic CD4⁺ cells are associated with target cell killing. We therefore investigated if our observed change in T cell cytotoxicity was associated with

**Fig. 1 | G9a/GLP inhibitors improve the cytotoxicity of TCR-engineered T cells.** **A** Workflow for enhancing engineered TCR-T cell therapy formation using epigenetic drugs and cytotoxicity validation in 2D and 3D. Created with BioRender.com. **B** Cytotoxicity of T cells towards HepG2-preS1 target cells, after 5 days of treatment with a drug panel from the Structural Genomics Consortium (SGC) Open Chemistry Networks platform (Open Chem Networks) targeting epigenetic factors. Data at 48 h after co-culture of T cells and target cells are shown. Data are shown as arbitrary units of fluorescence, corresponding to cytotoxicity, following manufacturer protocol for the CellTox assay. $N = 3$ biologically independent donors. **C** Engineered TCR-T cell cytotoxicity towards the target cell line after drug treatment in a 3D cytotoxicity assay. UNC0642-treated T cells resulted in a significant increase in target cell death compared to untreated T cells ($p = 0.0342$). $N = 3$ biologically independent donors. **D** Representative image of 2D cell cytotoxicity assay, tracked using xCelligence. Cell index was normalized to time when T cells were added, and T cells were added at a 1:1 effector: target ratio. Cell index at **E** 3.5-h, **F** 10-h, and **G** 17-h after T cell addition. $N = 3$ biologically independent donors. **H** Representative images showing live (green) and dead (red) cells in 3D cytotoxicity assay, with quantification of **I** normalized live target cell count after addition of untreated or UNC0642-treated TCR⁺ T cells, TCR⁻ mock-electroporated T cells (TCR⁻) and naïve T cells (no EP [electroporation]), normalized to no T cell controls and **J** number of invading T cells within the 3D matrix, normalized to no EP controls. TCR⁻ and no EP: $N = 3$ biologically independent donors; TCR⁺: $N = 6$ biologically independent donors. Data are analyzed with two-way ANOVA with matching within donors. All data are shown as mean ± SEM. *$P < 0.05$, **$P < 0.01$, ***$P < 0.001$, ****$P < 0.0001$, ns not significant. Source data are provided as a Source Data file.

different T cell subpopulations. UNC0642 treatment did not affect CD4⁺ and CD8⁺ proportions (Fig. 2A and Supplementary Fig. 2A, B). We also did not observe any changes in distinct CD4⁺ T cell subpopulations (Treg, Th1, Th2, and Th17; Supplementary Fig. 2A, C), CD4⁺ memory subtypes (Supplementary Fig. 2D, E) or CD8⁺ memory subtypes (Supplementary Fig. 2D, F).

Increased T cell cytotoxicity is also linked to terminal differentiation of effector T cells, and subsequent deterioration of T cell function, or T cell exhaustion[29]. We therefore investigated T cell activation and exhaustion after UNC0642 treatment (Fig. 2B, C). UNC0642 treatment resulted in a significant increase in granzyme B (GzmB) and small increase in Interleukin-2 (IL2) expression while there were no changes in Perforin (Perf), Interferon gamma (IFNγ) and Tumor Necrosis Factor alpha (TNF) expression (Fig. 2A). UNC0642 treatment also led to a small increase in the expression of Programmed Cell Death protein 1 (PD1) and CD39 but had no effect on Cytotoxic T-Lymphocyte Associated Protein 4 (CTLA4), Lymphocyte Activating Gene 3 (LAG3) or T-cell Immunoglobulin domain and Mucin domain 3 (TIM3) expression (Fig. 2C). Because CD4⁺ and CD8⁺ T cells have different functions in T cell cytotoxicity, and a mix of CD4⁺ and CD8⁺ cells are used in our assays, UNC0642 might differentially affect these subpopulations, we further assessed activation and exhaustion markers in these two populations (Fig. 2D, E). UNC0642 treatment increased GzmB expression dramatically in CD4⁺ and CD8⁺ T cells, while we observed a lesser increase in IFNγ, TNF and IL2 expression that was not significant due to donor-dependent variability (Fig. 2D). We observed a small increase in CTLA4 expression in CD4⁺ T cells, and small increases in PD1, CTLA4 and CD39 expression in CD8⁺ T cells (Fig. 2E).

T cell terminal differentiation and exhaustion are proposed to occur in a multi-step process induced by T cell activation, and expression of activation markers can overlap with exhaustion markers alone[29]. This is consistent with our observations, in which T cell cytotoxicity and effector markers (GzmB and IL2) were upregulated simultaneously with some markers of activation/exhaustion (PD1 and CD39). To simulate T cell activation and exhaustion more accurately, we investigated the effect of UNC0642 on CD4⁺ and CD8⁺ T cell GzmB, PD1, CTLA4, CD39, and TIM3 expression when TCR⁺ T cells were co-cultured with HepG2-preS1 target cells for up to 48 h, which is within the window of TCR⁺ T cell cytotoxicity (Supplementary Fig. 1E, F). Measurements were taken just after UNC0642 treatment/before TCR transfection (−48 h from co-culture), 24 h after transfection (−24 h), and 24 h or 48 h after co-culture with target cells (Fig. 2F, G and Supplementary Fig. 3B–E). GzmB expression in CD4⁺ and CD8⁺ T cells increased after TCR transfection, and increased further after co-culture with target cells, peaking at 24 h (Fig. 2F, G). UNC0642 treatment resulted in a higher expression of GzmB in both populations of T cells before (−48 h) and after TCR transfection (−24 h) (Fig. 2F, G). This increase was sustained in CD8⁺ T cells after 24 h of co-culture with target cells (Fig. 2G). PD1 expression levels were unchanged for CD4⁺ T cells, while they increased after

co-culture for CD8⁺ T cells (Supplementary Fig. 3A, B). However, UNC0642 treatment did not affect PD1 expression in these cells (Supplementary Fig. 3B). CTLA4 expression increased to a lesser degree in CD4⁺ and CD8⁺ T cells, and we observed a small increase in CTLA4 expression in UNC0642-treated CD4⁺ T cells before co-culture (Supplementary Fig. 3C). CD39 and TIM3 expression remained relatively unchanged for CD4⁺ T cells, while for CD8⁺ T cells expression increased after co-culture (Supplementary Fig. 3D, E). However, their expression after UNC0642 treatment was not dramatically increased in CD4⁺ or CD8⁺ T cells (Supplementary Fig. 3D, E).

Taken together, the data suggested that UNC0642 treatment increases T cell cytotoxicity by increasing GzmB expression, without modifying T cell subpopulations or T cell activation or exhaustion.

## The G9a/GLP inhibitor UNC0642 changes chemokine expression and cytotoxicity pathways at the transcriptional level

Because UNC0642 inhibits the epigenetic regulator G9a/GLP, to understand the impact of UNC0642 at the genetic level, we conducted a targeted genomics screen using the NanoString nCounter CAR-T characterization gene expression panel that assays 770 genes known to be involved in T cell activation, exhaustion, metabolism, persistence, toxicity and phenotype. UNC0642 treatment resulted in differential expression of various chemokines (upregulated: *CCL18, CCL23, CCL1, CXCL8, XCL1/2*, and downregulated: *CX3CR1*) as well as genes related to T cell costimulatory activity (upregulated: *CTLA4*) and exhaustion (upregulated: *FOXP3*) (Fig. 3A and Supplementary Fig. 4A). Increased transcription of *CCL18*, *CCL23*, and *CCL1* was validated by qPCR (Fig. 3B). Additionally, we determined by ChIP-qPCR that levels of H3K9me2, a key G9a/GLP target and regulator of transcription, at *CCL18* and *CCL23* were significantly decreased (Fig. 3C). Increase in *CCL18* expression at the translational level was confirmed by ELISA (Fig. 3D, E).

Using the NanoString nSolver analysis platform, we observed that UNC0642 treatment increased the expression of genes associated with the Th2 pathway, cytotoxicity, innate-like T cells, Th17 and Treg cell signaling and glycolysis (Fig. 3F). By contrast, the expression of genes related to the Th1 pathway and type II interferon signaling was decreased (Fig. 3F). These results are consistent with our flow cytometry data suggesting that UNC0642 treatment increases T cell cytotoxicity to a large degree and activation to a smaller degree (Fig. 2A, B). Changes to cellular metabolism were also suggested in the Nanostring data, hence the mitochondrial and glycolytic capacity of T cells after UNC0642 treatment was analyzed by extracellular flux. We found that UNC0642 treatment increased the maximum mitochondrial respiration and spare capacity of T cells (Fig. 3G), indicating that oxidative phosphorylation was increased. UNC0642 treatment did not affect the glycolytic profile of T cells (Fig. 3H). Exhausted T cells suppress mitochondrial respiration and glycolysis, while effector T cells rely mainly on aerobic glycolysis to produce energy[30]. We observe an increase in mitochondrial respiration

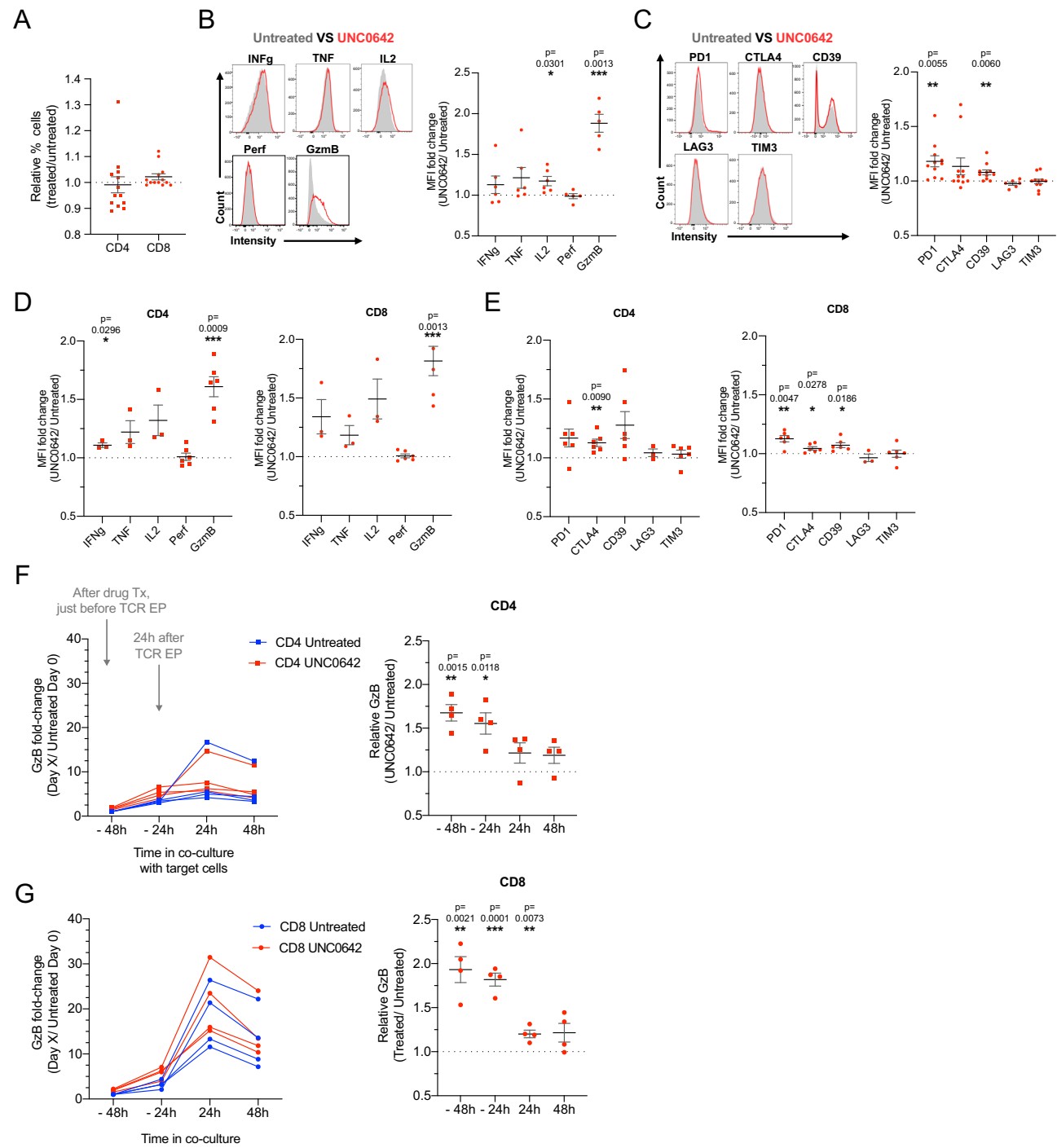

**Fig. 2 | UNC0642 treatment increases the cytotoxicity and of TCR-engineered T cells. A** Relative percentage of CD4+ and CD8+ T cells after UNC0642 treatment. Data are normalized to percentages in untreated paired samples. $N = 13$ biologically independent donors. **B** Representative plot of flow cytometry data and quantification of effector and activation proteins after UNC0642 treatment. Data are normalized to percentages in untreated paired samples. $N = 5$ biologically independent donors. **C** Representative plot of flow cytometry data and quantification of surface expression of proteins related to activation and exhaustion after UNC0642 treatment. Data are normalized to percentages in untreated paired samples. $N = 11$ biologically independent donors. **D** MFI of effector and activation markers in CD4+ and CD8+ T cells after UNC0642 treatment. Data are normalized to percentages in untreated paired samples. $N > 3$ biologically independent donors. **E** MFI of

activation and exhaustion markers in CD4+ and CD8+ T cells after UNC0642 treatment. Data are normalized to percentages in untreated paired samples. $N > 3$ biologically independent donors. **F** Granzyme B expression in untreated and UNC0642-treated TCR+ CD4+ T cells over time in co-culture with HepG2-preS1. Left panel: Data are normalized to untreated sample before TCR transfection at −48 h. Right panel: Data are normalized to untreated sample at each timepoint. $N = 4$ biologically independent donors. **G** Granzyme B expression in untreated and UNC0642-treated TCR+ CD8+ T cells over time in co-culture with HepG2-preS1. Data are analyzed with one-sample Wilcoxon signed rank test comparing against a hypothetical value of 1. $N = 4$ biologically independent donors. All data are shown as mean ± SEM. *$P < 0.05$, **$P < 0.01$, ***$P < 0.001$, ****$P < 0.0001$. Each dot represents a separate donor. Source data are provided as a Source Data file.

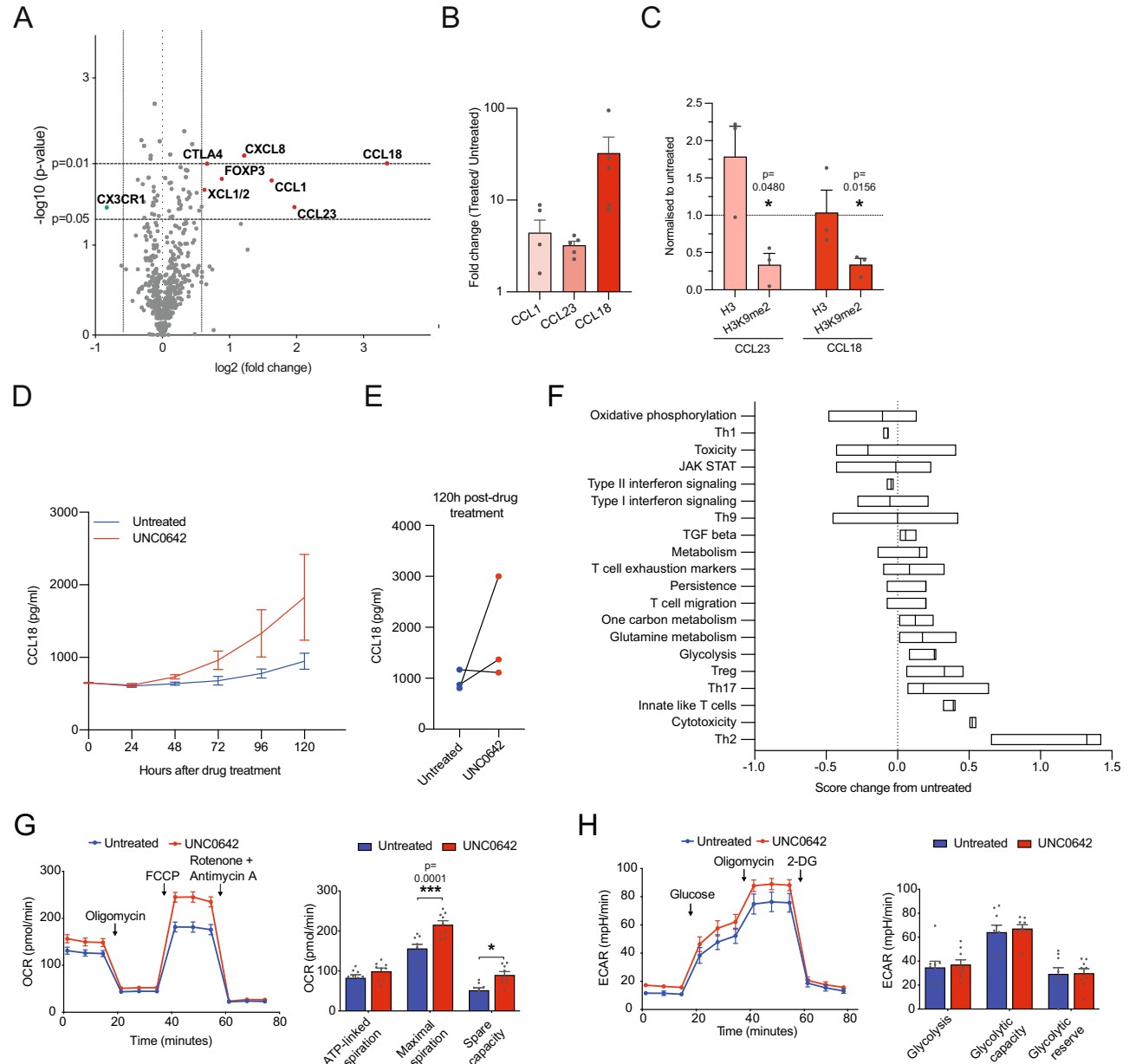

**Fig. 3 | Gene expression changes after UNC0642 treatment assayed using Nanostring CAR-T characterization panel. A** Volcano plot of changes in gene expression assayed with Nanostring CAR-T characterization panel for T cells after drug treatment. Genes with a significant ($p < 0.05$) increase of greater than 1.5 fold are in red, and less than 1.5 fold in green. Data are analyzed with two-sided t tests, $N = 3$ biologically independent donors. **B** qPCR validation of changes in gene expression after drug treatment. Data are represented as fold-change from untreated. Data are analyzed with two-sided one-sample t test against a hypothetical mean of 1, $N = 5$ biologically independent donors. **C** ChIP-qPCR validation of changes in H3K9me2 at *CCL23* and *CCL18* after drug treatment. Data are represented as fold-change from untreated. Data are analyzed with two-sided one-sample t test against a hypothetical mean of 1, $N = 3$ biologically independent donors. **D** Secreted CCL18 concentration after drug treatment assayed by ELISA.

$N = 3$ biologically independent donors. **E** Secreted CCL18 concentration at 120 h after drug treatment. Data for individual donors are shown. **F** Change in pathway scores after drug treatment. Pathway scores were defined using nSolver Advanced analysis software. Data are from 3 different donor T cells. **G** Oxygen consumption rate (OCR) of untreated and UNC0642-treated T cells. ATP-linked respiration, maximal respiration and spare capacity were calculated from the OCR plot. Data are analyzed with two-way ANOVA with multiple comparisons, $N = 3$ biologically independent donors. **H** Extracellular acidification rate (ECAR) for untreated and UNC0642-treated T cells. Glycolysis, glycolytic capacity and glycolytic reserve were calculated from the ECAR plot. Data are analyzed with two-way ANOVA with multiple comparisons, $N = 3$ biologically independent donors. All data are shown as mean ± SEM. *$P < 0.05$, **$P < 0.01$, ***$P < 0.001$, ****$P < 0.0001$. Source data are provided as a Source Data file.

and no changes in glycolysis after UNC0642 treatment, which is consistent with our flow cytometry data showing that UNC0642 treatment increased the expression of effector T cell markers, but not exhaustion markers (Fig. 2A, B and Supplementary Fig. 3B–E). Overall, the data indicate that UNC0642 is affecting the expression of specific genes related to T cell effector function, such as cytotoxicity, activation, and mitochondrial respiration.

**Proteins associated with T cell activation and cytotoxicity are increased after UNC0642 treatment**

To fully characterize the effect of UNC0642 on T cells and to account for possible post-transcriptional modifications, we performed a high-throughput proteomic screen with quantitative proteomics using tandem mass tags (TMT). Briefly, T cells were treated with UNC0642, and samples from treated and untreated T cells were collected every

24 h until day 5. Protein samples were labeled using distinct TMT tags and the protein abundance was determined by comparing treated and untreated T cells over time. A total of 4918 proteins were identified and quantified; of these, 2243 (~45%) were identified on all 5 days. Only a small number of proteins exhibited changes in abundance (log2 fold change ±0.5, $P \leq 0.05$) over time in UNC0642-treated T cells compared to untreated T cells (6, 22, 10, 15, and 7 on days 1–5, respectively) (days 1–4: Supplementary Fig. 5A–D and day 5: Fig. 4A). On day 1, we observed an increase in proteins involved in gene regulation such as zinc finger protein 587B (ZNF587B) and primase and DNA directed polymerase (PRIMPOL), as well as lysosomal protein transmembrane 5 (LAPTM5), Interleukin 32 (IL32) and Serine Incorporator 1 (SERINC1) (Supplementary Fig. 5A). On Day 2, we observed increases in proteins involved in post-translational protein modification and regulation such as DnaJ Heat Shock Protein Family (Hsp40) Member C10 (DNAJC10), G Protein-Coupled Receptor Associated Sorting Protein 2 (GPRASP2), Ankyrin Repeat Domain 28 (ANKRD28), and ST3 Beta-Galactoside Alpha-2,3-Sialyltransferase 3 (ST3GAL3); as well as increases in proteins involved in gene expression regulation such as TMF1 Regulated Nuclear Protein 1 (TRNP1) and CCHC-Type Zinc Finger Nucleic Acid Binding Protein (CNBP; Supplementary Fig. 5B). On Day 3, there was an increase in Mediator Complex Subunit 1 (MED1), involved in gene transcription, and Dishevelled Associated Activator Of Morphogenesis 1 (DAAM1), involved in Wnt and Rho signalling, as well as an increase in proteins associated with the lysosomal pathway – LAPTM5, NPC Intracellular Cholesterol Transporter 2 (NPC2) and Serpin Family B Member 6 (SERPINB6) (Supplementary Fig. 5C). On day 4, we observed an enrichment for granzyme B, proteins involved in immune cell signalling – Major Histocompatibility Complex, Class II, DR Alpha (HLA-DR), HCLS1 Binding Protein 3 (HS1BP3), Immunoglobulin Lambda Constant 2 (IGLC2), Immunoglobulin Lambda Like Polypeptide 5 (IGLL5; Supplementary Fig. 5D).

On the last day of UNC0642 treatment, granzymes B and H (GZMB, GZMH), lysosomal-associated membrane protein 1 (LAMP-1), macrophage-capping protein (CAPG) and epididymal secretory protein E1 (NCP2) were significantly upregulated, while only DEAD-box helicase 54 (DDX54) was downregulated (Fig. 4A). GO enrichment of the top 50 upregulated proteins included functions related to processes including immune response, cell activation and cell killing, while GO enrichment of downregulated proteins showed only a small enrichment in metabolic processes (Fig. 4B). We validated the changes in expression of GZMB and GZMM, and confirmed an increase after UNC0642 treatment, and perforin expression, which did not increase significantly (Fig. 4C). Notably, the degree of change detected by flow cytometry matched that detected in the proteomics analysis, providing additional confidence in the validity of the hits (Fig. 4D).

To identify trends in protein abundance over the 5 days of UNC0642 treatment, unsupervised k-means clustering was applied to the 2,243 proteins identified across all 5 days, resulting in 10 different clusters with similar abundance profiles during UNC0642 treatment (Fig. 4E and Supplementary Fig. 5E). Majority of proteins were classified in clusters with a transient change in abundance (clusters 5, 6, 7, and 10, Supplementary Fig. 5E) or no change over the 5 days (clusters 1, 2, 3, and 9, Supplementary Fig. 5E). By contrast, proteins in cluster 8 had an increased abundance while proteins in cluster 4 had a slight decrease in abundance. Proteins that were upregulated at day 5 (Fig. 4A) were identified in cluster 8 (GZMB, CAPG, GZMM, NCP2, GZMH) (Fig. 4E), indicating a sustained increase in the expression of these proteins during treatment. GO analysis of the proteins in cluster 8 showed enrichment in immune response, immune cell activation and effector processes (Fig. 4F), while proteins with slightly decreased expression in cluster 4 were enriched in metabolic processes (Fig. 4F). Overall, the data indicate that UNC0642 treatment affects T cell behavior at the protein level, with a sustained increase in proteins associated with T cell activation and cytotoxicity. Additionally, the

data suggest that UNC0642 treatment triggers early changes in proteins involved in gene expression and post-translational protein modification and regulation, and later changes in proteins in the lysosomal pathway and T cell signalling.

## UNC0642 treatment increases engineered TCR⁺ T cell antitumor activity in an orthotopic mouse model of HCC

Given the encouraging data for engineered T cells in vitro, we next examined the ability of UNC0642 treatment to increase the cytotoxicity of transient engineered TCR⁺ T cells in vivo. We established an orthotopic xenograft model in NSG mice by intrahepatic injection of HepG2-2.2.15 target cells expressing luciferase, to better recapitulate the T cell targeting and cytotoxicity to the liver than in a subcutaneous xenograft model. $1 \times 10^6$ untreated or UNC0642-treated TCR⁺ or mock electroporated TCR⁻ T cells were introduced intravenously on days 7, 9, 11, 13, 15, 18, 20, and 22 post-tumor cell injection (Fig. 5A) and the tumor burden was monitored by in vivo bioluminescence imaging (IVIS) (Fig. 5B). Tumor growth was comparable between untreated mice (Fig. 5C, green solid line) and mice receiving untreated TCR⁻ T cells (Fig. 5C, blue dashed line). Remarkably, UNC0642-treated TCR⁻ T cells restricted tumor growth for the first 20 days of observation (Fig. 5C, red dashed line), before tumor regression was observed at day 22. Untreated TCR⁺ T cells restricted tumor growth up till day 18 (Fig. 5C, blue solid line), before tumor regression was observed after day 20. In contrast, tumor growth in mice receiving UNC0642-treated TCR⁺ T cells was significantly decreased throughout the 22 days of observation (Fig. 5C, red solid line). We also observed a trend where mice receiving UNC0642-treated TCR⁺ or TCR⁻ T cells had less weight loss than mice receiving untreated TCR⁺ or TCR⁻ T cells or no T cells (Fig. 5D). Histological analysis confirmed that injected TCR⁺ T cells (CD3⁺ staining) were appropriately targeting the liver tumor in mice receiving untreated or UNC0642-treated TCR⁺ T cells (Fig. 5E). Automated segmentation for CD3⁺ regions in multiple liver histological sections from separate mice showed that there was an increase in T cell presence in the livers of mice receiving UNC0642-treated TCR + T cells (Fig. 5E). Overall, the data suggest that UNC0642 treatment improves engineered TCR⁺ T cell anti-tumor activity, likely by improving T cell persistence and cytotoxicity in the liver.

## UNC0642 treatment increases the cytotoxicity of patient-derived engineered TCR-T cells

Engineered TCR⁺ T cells used in the clinic are usually generated by modifying T cells obtained from patients, which may have impaired cytotoxicity due to prolonged tumor progression or chemotherapy[31]. We therefore investigated the ability of UNC0642 treatment to improve engineered TCR⁺ T cells generated from patients diagnosed with HBV-related HCC[32]. T cells were isolated from two HBV-related HCC patients and transduced, before carrying out 2D and 3D cytotoxicity assays (Fig. 6A). At the time of patient blood withdrawal, Patient 1 had a lymphocyte count within normal range, while Patient 2 had a low lymphocyte count (Fig. 6B). However, T cells derived from both Patient 1 and 2 were able to expand and undergo TCR transduction ex vivo successfully[32]. UNC0642 treatment increased expression of granzymes B and M and perforin (Fig. 6B and Supplementary Fig. 6A). In Patient 1, engineered TCR⁺ T cells were much more efficient at target cell killing than non-transduced TCR⁻ T cells (compare pink and blue lines in Fig. 6C top panel, TCR⁻ and TCR⁺ blue points in Fig. 6D top panel), and UNC0642 treatment did not result in any significant changes in target cell lysis compared with the untreated TCR⁺ cells (Fig. 6C, D, top panel). By contrast, in Patient 2 TCR⁺ T cells did not induce target cell death as effectively, both in 2D (compare blue lines in top and bottom panels in Fig. 6C) and 3D (compare untreated TCR⁻ T cells to TCR⁺ T cells for top and bottom panels in Fig. 6D). However, for Patient 2, UNC0642 treatment of TCR⁺ T cells resulted in a dramatic increase in target cell death compared to untreated TCR⁺ T cells

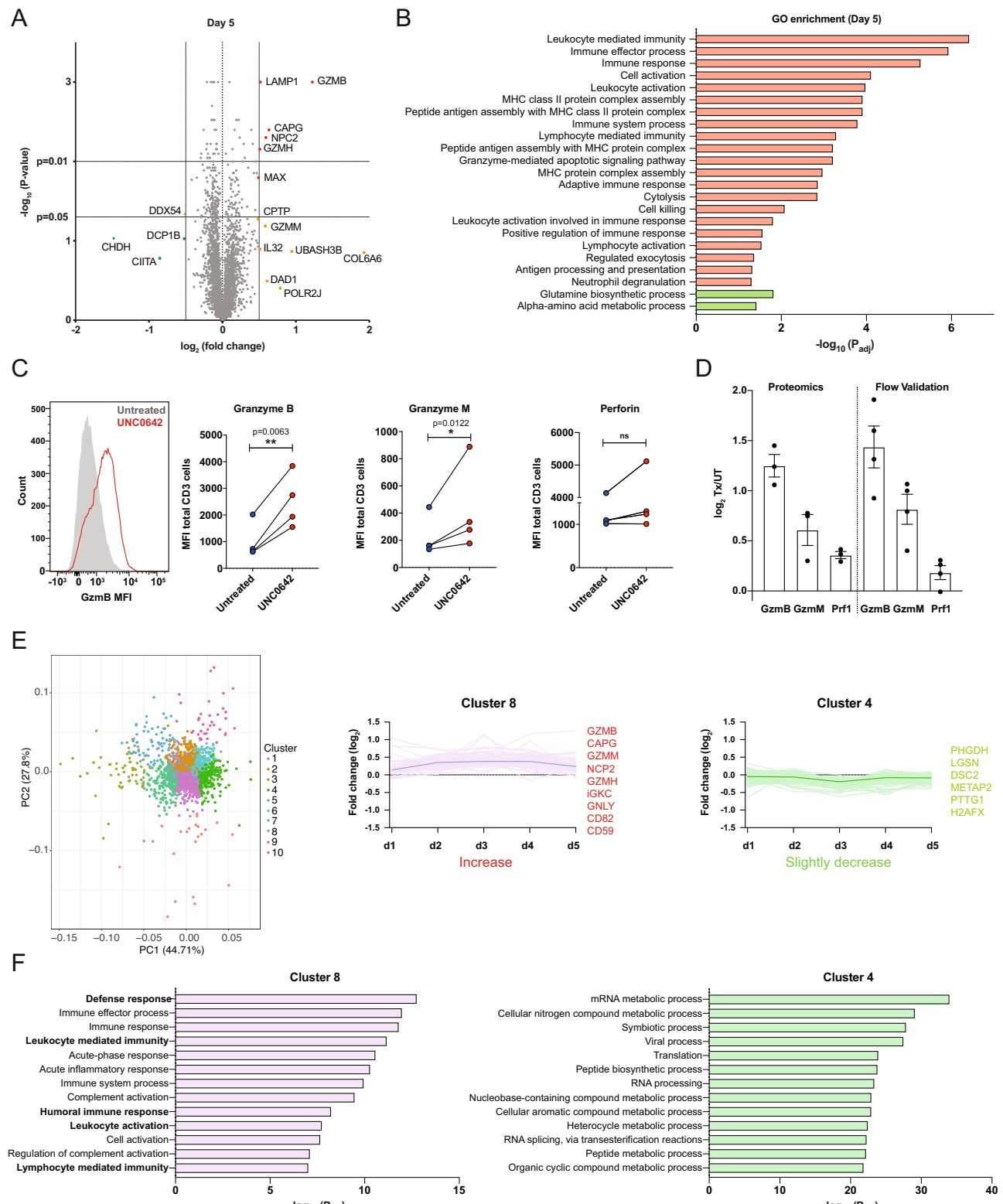

**Fig. 4 | Changes in protein abundance in T cells after UNC0642 treatment assayed using TMT labeling. A** Changes in protein abundance of T cells after 5 days of UNC0642 treatment. Data are analyzed with two-sided $t$ tests, $N = 3$ biologically independent donors. **B** Change in pathway scores after drug treatment of the differentially abundance proteins. Pathway scores and significance were identified using Panther classification system. Upregulated pathways are indicated in red and downregulated pathways are indicated in green. **C** Flow cytometry validation of changes in protein abundance in GZMB, GZMM and PRF1 after drug treatment. Data are non-normalized MFI reads and are analyzed with two-sided paired t-test, $N = 4$ biologically independent donors. **D** Comparison of protein expression assayed in by proteomics as in **A** and flow cytometry as in **C**. $N = 3$ donors for proteomics; 4 donors for flow cytometry. Data are represented as fold-change from untreated paired samples and are shown as mean ± SEM. **E** Proteins grouped by changes in abundance over time using k-means clustering, represented on a PCA plot (left) and on a plot showing fold-change over time (right). All data points are plotted and dark lines indicate mean for each group. **F** Change in pathway scores in the Cluster 8 (proteins increasing with the UNC0642 treatment) and 4 (proteins decreasing with UNC0642 treatment). *$P < 0.05$, **$P < 0.01$. Source data are provided as a Source Data file.

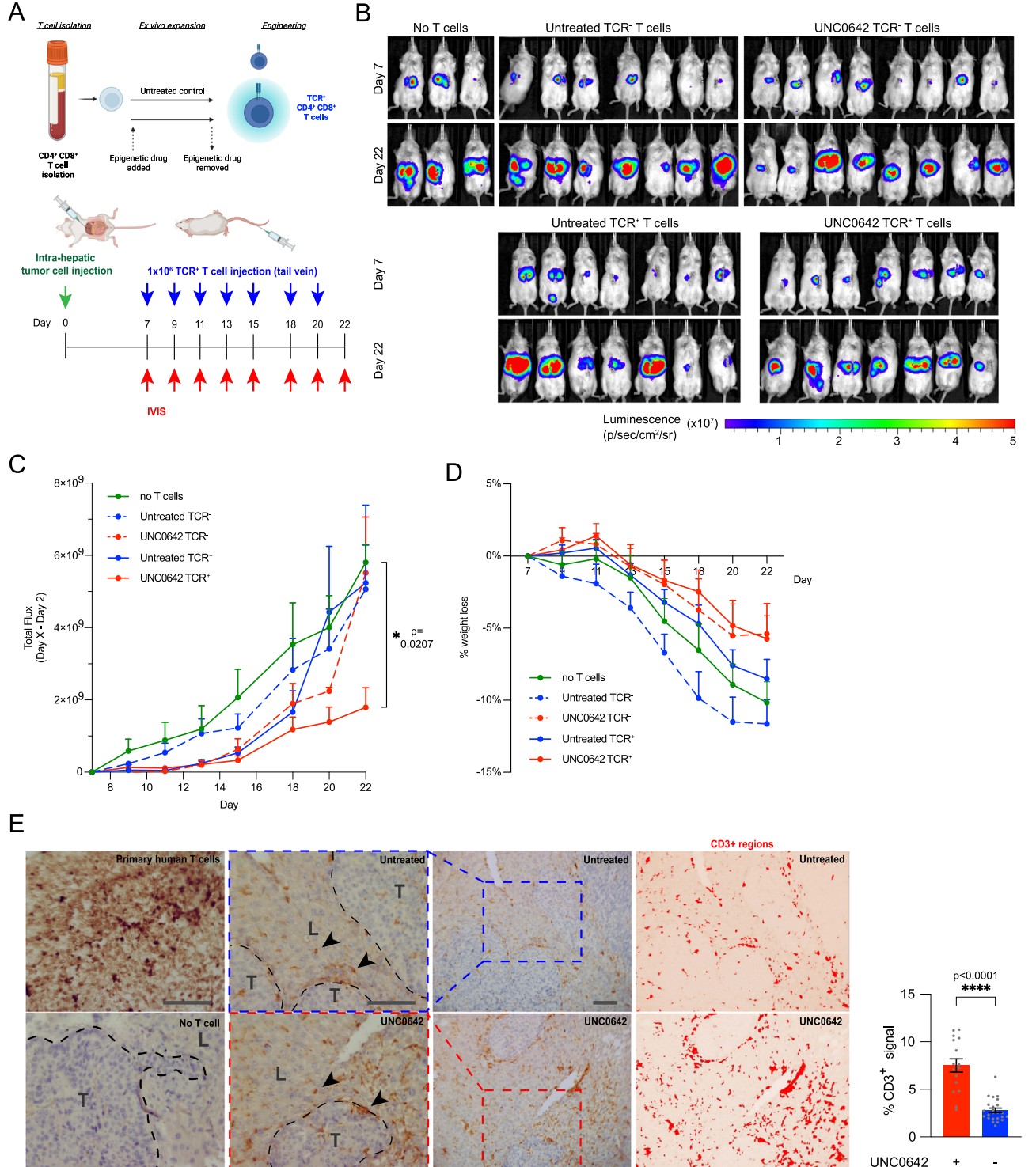

**Fig. 5 | UNC0642 treatment improves the ani-tumor activity in vivo. A** Overall design of in vivo experiment. 2 M HepG2-2.2.15-luc cells were injected intra-hepatically at the start of the experiment. At day 7, mice were allocated into 5 groups and 1 M of untreated or UNC0642-treated TCR⁺ or TCR⁻ T cells were injected intra-venously. Tumor volume was monitors by IVIS imaging and T cell injections were repeated every 2 days until day 21 post-tumor injection. Created with BioRender.com. **B** Images taken using the in vivo bioluminescence imaging system at day 7 and at day 21 post-tumor injection. All mice analyzed are shown. **C** Tumor volume was tracked using bioluminescence and shown as total photon flux (p/s) relative to day 7 post-tumor injection over time. Data at day 21 are analyzed with one-way ANOVA with multiple comparisons. $N = 3$ for no T cell control mice; 8 for UNC0642-treated TCR⁻ T cell mice; 7 for all other conditions. **D** Percentage loss of body weight. Data are analyzed with one-way ANOVA with multiple comparisons, $N > 3$ mice per condition, as shown in **B**. **E** Immunohistochemical images of liver sections stained for CD3 to identify T cells, with normal liver (L) and tumor (T) areas indicated and arrowheads point to T cells. Automatically segmented images of CD3 + regions in liver sections stained for CD3. Primary human T cells serve as positive staining control and no T cell sample as negative staining control. Scale bar represents 50um. Quantification of CD3⁺ regions in automatically segmented liver sections from tumors treated with untreated and UNC0642-treated TCR⁺ T cells. Data are analyzed with two-sided Mann–Whitney test, $N > 4$ biologically independent mice, 4 analyzed sections per mouse. All data are shown as mean ± SEM. *$P < 0.05$, **$P < 0.01$, ***$P < 0.001$, ****$P < 0.0001$. Source data are provided as a Source Data file.

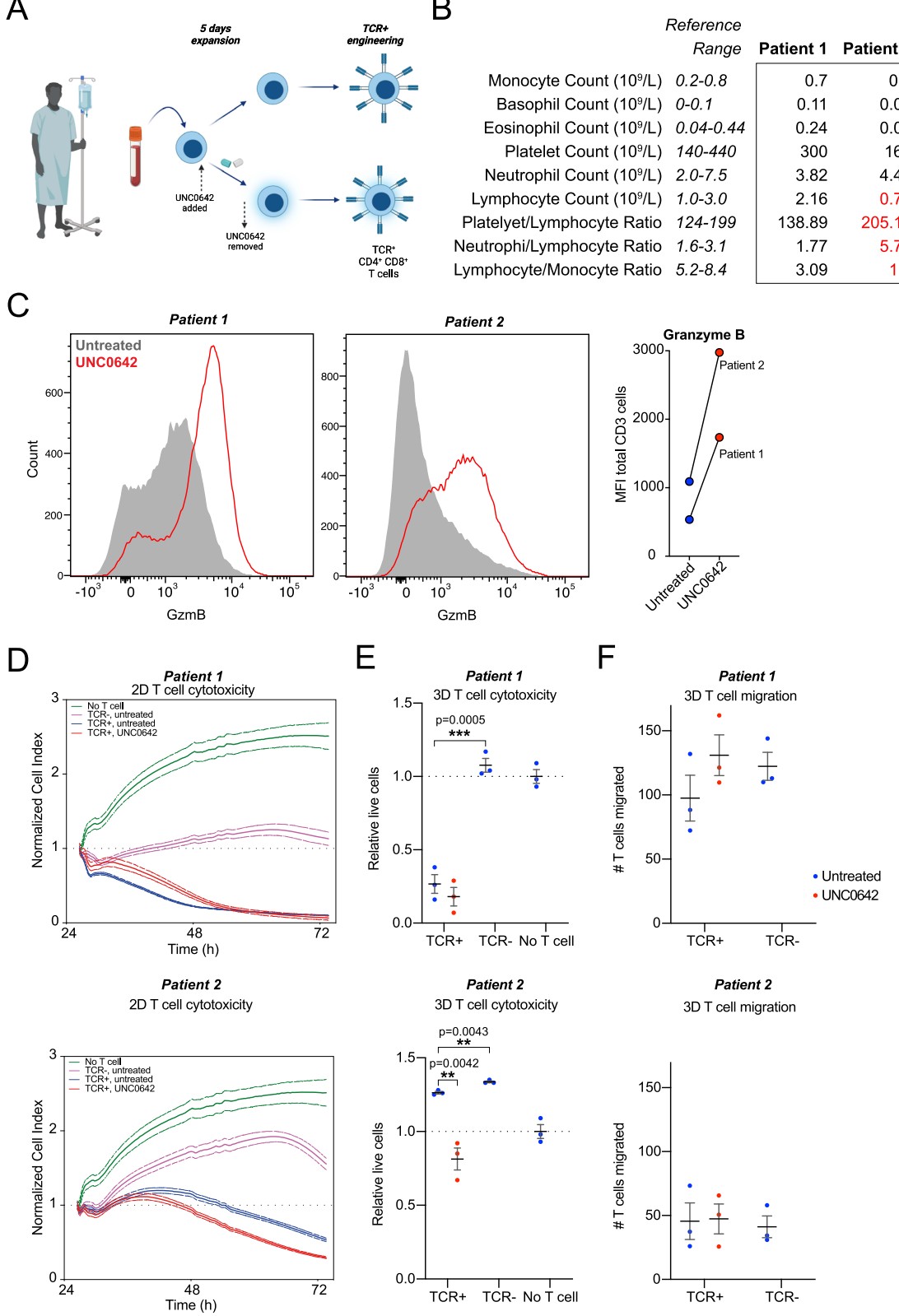

**Fig. 6 | UNC0642 treatment increases the cytotoxicity of patients T cells.**
**A** Workflow for UNC0642 treatment of engineered patient TCR⁺ T cell therapy formation and validation. Created with BioRender.com. **B** Complete blood count for Patients 1 and 2. Patient values outside of reference range are in red. **C** Granzyme B expression levels in patient T cells after UNC0642 treatment. Each data point represents a single experimental replicate for the same donor. **D** 2D target cell killing assay for Patient 1 (top panel) and Patient 2 (bottom panel). Data are

normalized to cell index at time of T cell addition, and are represented as mean with standard error of the mean over time. **E** 3D target cell killing assay for Patient 1 and Patient 2. Data are normalized to no T cell target cell numbers. Data are analyzed with one-way ANOVA. $N = 3$ experimental replicates for the same donor. **F** T cell migration in 3D target cell killing assay for Patient 1 and Patient 2. $N = 3$ experimental replicates for the same donor. $*P < 0.05$, $**P < 0.01$, $***P < 0.001$, $****P < 0.0001$. Source data are provided as a Source Data file.

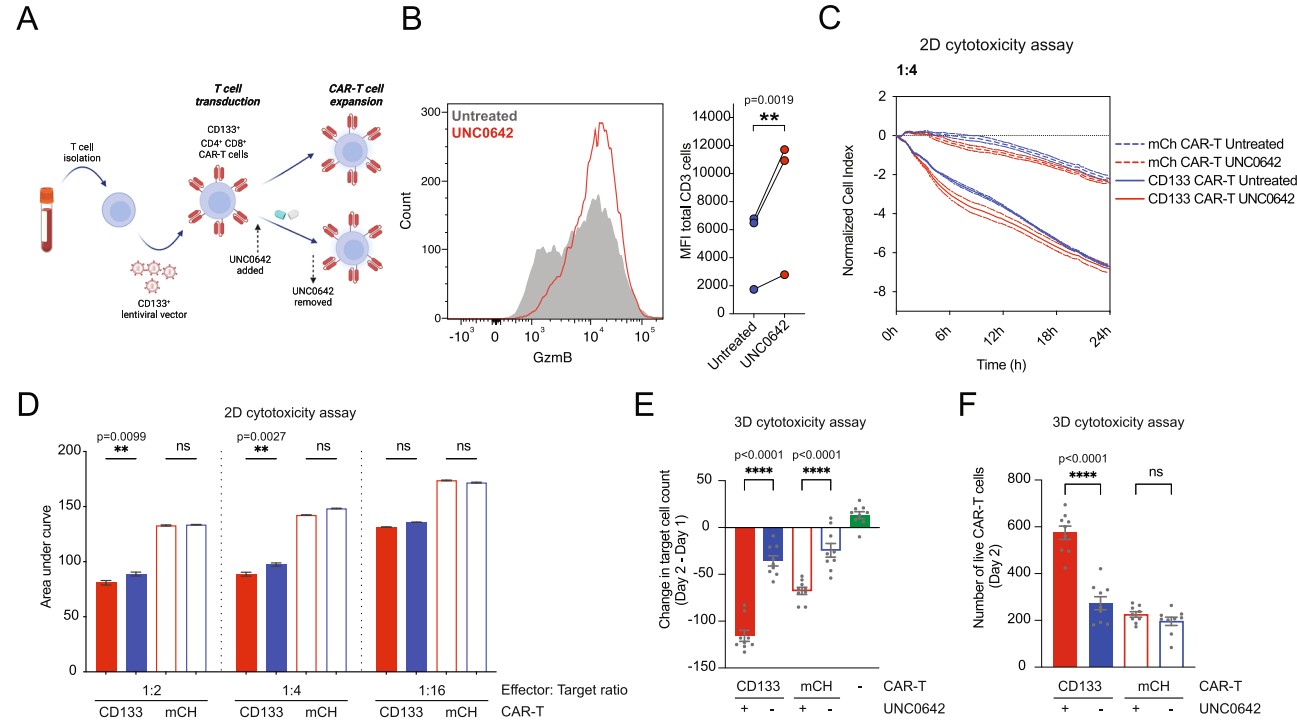

**Fig. 7 | UNC0642 treatment increases engineered CAR-T cell cytotoxicity.**
**A** Workflow for CD133⁺ CAR-T production and UNC0642 treatment. Created with
BioRender.com. **B** Granzyme B expression levels in CAR-T cells after UNC0642
treatment (*n* = 3 donors). **C** Example normalized cell index for 2D target cell killing
assay for untreated and UNC0642-treated CD133⁺ and mCH (CD133⁻) CAR-T cells.
CAR-T cells were added to target cells at a 1:4 effector: target ratio (*n* = 3 donors).
**D** Area under the curve for normalized cell index as show in **C** for untreated and
UNC0642-treated CD133⁺ and mCH CAR-T cells at 1:2, 1:4 and 1:16 effector:target
ratios. Data are analyzed with one-way ANOVA with multiple comparisons (*n* = 3

donors). **E** 3D target cell killing assay for untreated and UNC0642-treated CD133⁺
and mCH CAR-T cells at day 2 after CAR-T addition. Data are normalized to no T cell
target cell numbers. Data are analyzed with one-way ANOVA with multiple com-
parisons. *N* = 9 analyzed ROI regions from three biologically independent donors.
**F** CAR-T cell migration in 3D target cell killing assay for untreated and UNC0642-
treated CD133⁺ and mCH CAR-T cells at day 2. Data are analyzed with one-way
ANOVA with multiple comparisons. *N* = 9 analyzed ROI regions from three biolo-
gically independent donors. All data are shown as mean ± SEM. *P < 0.05, **P < 0.01,
***P < 0.001, ****P < 0.0001. Source data are provided as a Source Data file.

(bottom panel, Fig. 6C, D). One possible explanation for the lack of
improvement in UNC0642-treated TCR⁺ T cells in Patient 1 might be
that because TCR engineering was so effective in this patient, further
improvements to T cell cytotoxicity due to UNC0642 treatment might
be minimal. UNC0642 treatment did not affect T cell migration
(Fig. 6F), however T cell migration into the extracellular matrix was
higher for Patient 1 than Patient 2 (Fig. 6F, compare top and bottom
panel). It is worth noting that in a previous clinical study, Patient 1
responded to adoptive immunotherapy while Patient 2 did not[32].
Taken together, the data suggests that lymphocytes from Patient 2
might have impaired function that resulted in less effective engineered
TCR⁺ T cells, which improved after UNC0642 treatment. This suggests
that UNC0642 treatment may be particularly useful for improving the
cytotoxicity of TCR⁺ T cells engineered from patients with compro-
mised lymphocytes.

## UNC0642 treatment increases the cytotoxicity of stable engi-neered CAR-T cells

T cells stably-expressing chimeric antigen receptors (CARs) are also an
important adoptive cell therapy product with synthetic immunor-
eceptors that specifically target cancer cells and contain co-
stimulatory receptors such as CD28 or 4-1BB that promote effector
cell function, proliferation and persistence[33]. To understand if
UNC0642 treatment could improve CAR-T cell cytotoxicity, we
investigated the effect of UNC0642 on CAR-T cells targeting CD133
(Fig. 7A), expressed by cancer stem cells originating from various
epithelial cells[34], including the liver cancer cell line Hep3B (99.6 %
CD133⁺; Supplementary Fig. 6B). UNC0642 treatment significantly

increased the production of granzymes B and M and perforin in CAR-T
cells (Fig. 7B and Supplementary Fig. 6C). In functional 2D cytotoxicity
assays, untreated CD133 CAR-T cells exhibited remarkable target cell
lysis, even at lower effector: target ratios (Supplementary Figure 6D-E)
and UNC0642 treatment improved CAR-T cell cytotoxicity slightly at
1:2 and 1:4 effector:target ratios (Fig. 7C, D). As an additional control,
untreated, and UNC0642-treated non-targetting CD19-directed CAR-T
(henceforth referred to as mCH CAR-T) were used. mCH CAR-T
induced non-specific target cell death, but was not as efficient as CD133
CAR-T cells (Fig. 7C, D and Supplementary Fig. 6D, E). UNC0642
treatment did not increase non-specific target cell death due to mCH
CAR-T in 2D (Fig. 7C, D and Supplementary Fig. 6D, E). In 3D cyto-
toxicity assays, we observed significant target cell death both with
untreated CD133 CAR-T cells and untreated mCH CAR-T cells (Fig. 7E).
Target cell death was increased with UNC0642 treatment, and this
increase was very marked for CD133 CAR-T cells (Fig. 7E). Both
untreated CD133 and mCH CAR-T migrated into the extracellular
matrix of the 3D cytotoxicity assay, and we observed an increase in
CAR-T migration for UNC0642-treated CD133 CAR-T (Fig. 7F). The data
suggest that even though CAR-T cells already express co-stimulatory
domains for improved effector function, UNC0642 treatment is able to
improve the cytotoxicity of these cells. And at least for CD133 CAR-T
cells, UNC0642 treatment improves CAR-T cell migration in 3D.

## UNC0642 treatment increases the cytotoxicity of NK cells

The data thus far suggests that UNC0642 increases both non-
engineered and engineered T cell cytotoxicity. To assess whether
UNC0642 treatment might also increase granzyme expression and

cytotoxicity in other cytotoxic cell types, we evaluated the effect of UNC0642 on NK cells. NK cells have a critical function in immune activation against abnormal cells, including cancer cells and perform an essential function in tumor immunosurveillance[35]. NK cells were isolated from 4 different donors and treated for 5 days with UNC0642 (Fig. 8A). UNC0642 treatment significantly increased the levels of granzymes B and M and perforin in NK cells from all four donors compared to the untreated NK cells (Fig. 8B and Supplementary Fig. 6F). Accordingly, UNC0642 treatment increased NK cell cytotoxicity in the 2D cytotoxic assay, with an observable effect at 1:2, 1:4, and 1:8 effector:target ratios (Fig. 8C, D). In the 3D cytotoxicity assay, we observed a small decrease in target cell growth with untreated NK cells, and a significant decrease with UNC0642-treated NK cells (Fig. 8E). It should be noted that NK cells were not engineered for target cell specificity, therefore the observed cell death is modest compared to that of engineered TCR⁺ or CAR-T cells. Overall, UNC0642 treatment appears to increase granzyme expression and cytotoxicity of multiple engineered and innate cytotoxic cell types.

## Discussion

Transient engineered TCR⁺ T cells are a valuable and effective antitumor therapy, while also carrying low to no risk of cytokine release syndrome or insertional mutagenesis. However, their transient nature does mean they have a shorter window of antitumor activity and multiple infusions are required, and so there is a need to improve their efficacy further to minimize patient cost and discomfort. By screening small molecules that regulate epigenetic players, we have exploited epigenetic T cell regulation to improve the cytotoxicity of transient engineered TCR⁺ T cells. We found that low-dose inhibition of G9a/GLP by UNC0642 treatment of T cells during their ex vivo expansion phase resulted in increased T cell cytotoxicity in multiple in vitro and in vivo assays. This increase in cytotoxicity is likely due to an increase in granzyme expression, especially GZMB. Additionally, we found an increase in T cells in the livers of mice treated with UNC0642-treated TCR⁺ T cells, suggesting that UNC0642 treatment might also facilitate increased T cell persistence. Interestingly, G9a/GLP inhibition during cell expansion protocols increased the granzyme expression and cytotoxic capability of TCR-engineered T cells (from healthy and patient donors), CAR-T cells and NK cells, suggesting that G9a/GLP is involved in granzyme expression and cytotoxicity across multiple cytotoxic engineered and non-engineered immune cell types.

G9a/GLP has been reported to regulate immune cell differentiation and functions[28], such as type II cytokine production[36], and Th17/Treg cell differentiation[37]. In accordance with this, we found changes in genes related to Th2, Th17 and Treg cells in our targeted genomics screen. However, we did not detect any changes to the T cell subpopulations after UNC0642 treatment during the expansion phase. It is likely that while UNC0642 treatment at 1.25 μM for 5 days was sufficient to reduce H3K9me2 levels by approximately 50%, this dose was insufficient to result in the re-programming of T cells observed in other studies in where knock-out models or higher pharmacological doses were used. Additionally, most studies investigated G9a/GLP in naïve T cell differentiation, while our study involves differentiated T cells, where the epigenetic landscape is different. In differentiated immune cells, G9a has been implicated in CD8⁺ effector memory cell persistence[38], as well as macrophage tolerance to chronic endotoxin infection[39–41]. Similarly, our data shows that the antitumor activity of CD4⁺/CD8⁺ T cells is improved following G9a/GLP inhibition, due to an increase in T cell cytotoxicity, via an increase in granzyme production, rather than T cell targeting. Notably, the increase in cytotoxicity was not accompanied by characteristic features of T cell exhaustion, such as a decrease in cell proliferation, expression of T cell exhaustion markers, or decreases in mitochondrial respiration and glycolytic capability. Our findings also highlight the importance of using a range of functional assays, such as 3D in vitro models and orthotopic in vivo

models to evaluate T cell effector function, as the expression of activation and exhaustion markers alone was not always indicative of T cell function.

Both genomic and proteomic analyses found increases in T cell activation and cytotoxicity, although the same genes were not identified. Discordance between genomic and proteomic profiles are common[42]; however, comparing differentially expressed genes and proteins provides greater confidence in the data[43]. Differences could be due to the read-outs used and changes at the post-transcriptional level. We found that UNC0642 treatment predominantly led to an upregulation of gene or protein levels, which is consistent with the canonical role of G9a/GLP as an epigenetic repressor. Previous studies where G9a was silenced in germ cells found only eight genes upregulated by more than 2-fold, despite significant loss of H3K9me2/1[44]. We observed a similar phenomenon, where G9a reduction by UNC0642 treatment affected only a handful of genes and proteins. G9a/GLP has also been reported to regulate methylation of non-histone targets, including G9a itself[28]; hence changes in cell behavior may be induced by changes in protein interactions rather than through changes at the genetic or protein level. We found significant changes in chemokine expression after UNC0642 in our targeted genomics screen, most significantly of which was CCL18. CCL18 is a chemoattractant for naïve T cells, T and B lymphocytes and NK cells[45]. CCL18 is also predominantly produced by M2 macrophages and TAMs, and can induce the differentiation of effector T cells into Treg cells[46]. CCL18 can also promote cancer cell invasion by inducing epithelial-mesenchymal transition (EMT)[47–51]. In our study, Treg cell subpopulations were unchanged following UNC0642 treatment, and we did not observe an increase in the tumor invasive front in our in vivo model following the transfer of UNC0642-treated T cells. Given that post-translational modifications can significantly modify chemokine activity[52] and their effects can be highly context-dependent, further research is needed into the function of CCL18 and other cytokines in T cell cytotoxicity and their regulation by G9a/GLP.

Increases in immune effector process found in our proteomic screen and an upregulation in proteins involved in the granzyme-mediated apoptotic signalling pathway. We observed early changes in proteins involved in gene expression, followed by changes in protein post-translational modifications, followed by changes in proteins involved in the lysosomal pathway, T cell signalling and granzyme production. The data suggests that UNC0642 is acting upon the granzyme production pathway by triggering a molecular cascade, starting with gene expression. Blimp1/G9a has been suggested to negatively regulate Granzyme B expression[38]. Although changes in Blimp1 were not found in our study, our data suggests a similar function of G9a/GLP in suppressing granzyme expression in effector T cells. Further studies involving the proteins we found differentially regulated in our screen and possible interactions with Blimp1 are warranted.

We demonstrated that G9a/GLP improved cytotoxicity in a range of cytotoxic cell types – TCR-engineered T cells from healthy and patient donors; CAR-T cells, as well as NK cells. This suggests that G9a/GLP-regulated H3K9me2 may be involved in a general mechanism that underlies the repression of cytotoxicity-related genes in terminally differentiated T cells and NK cells. Consistent with this, a general consensus is emerging whereby immune cell differentiation is associated with progressive increases in epigenetic repressive markers, such as H3K9me2 or H3K9me3[53]. Epigenetic regulation results in heritable changes in T cell behavior; however, we did not evaluate long-term T cell cytotoxicity and epigenetic changes. Additionally, it would be interesting to see if UNC0642 treatment has a similar effect on other cytolytic immune cell types, such as lymphokine-activated killer (LAK) cells.

In conclusion, we show that the use of a small molecule inhibitor of an epigenetic regulator, G9a/GLP, during ex vivo expansion is a convenient and effective method of improving the cytotoxicity of transiently engineered TCR-T cells, as well as other cytotoxic cell types

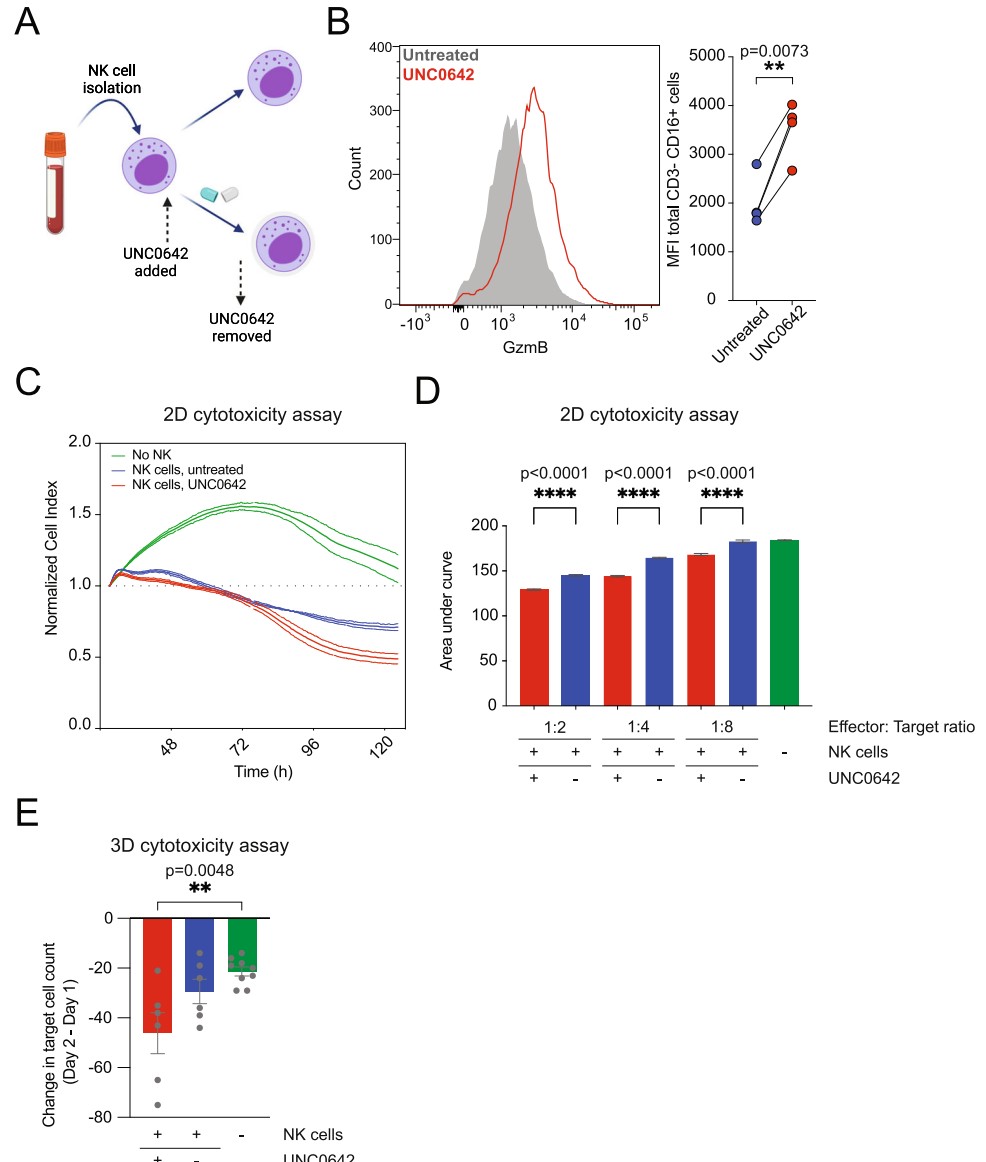

**Fig. 8 | UNC0642 increases NK cell cytotoxicity. A** Workflow for NK isolation and UNC0642 treatment during NK cell expansion. Created with BioRender.com. **B** Granzyme B expression levels in NK cells after UNC0642 treatment (*n* = 3 donors). **C** Example normalized cell index for 2D cytotoxicity assay for untreated and UNC0642-treated NK cells. **D** Area under the curve for the normalized cell index for the 2D cytotoxicity assay as shown in example plot in **C**. Data for untreated and UNC0642-treated NK cells at 1:2, 1:4 and 1:8 effector:target ratio are shown. Data are analyzed with one-way ANOVA with multiple comparisons. (*n* = 3 donors). **E** Normalized live target cell count for the 3D cytotoxicity assay for target cells after addition of untreated or UNC0642-treated NK cells (*n* = 3 donors, each data point represents an analyzed ROI). Data are analyzed with one-way ANOVA with multiple comparisons. Data shown are for four different donors. All data are shown as mean ± SEM. *$P < 0.05$, **$P < 0.01$, ***$P < 0.001$, ****$P < 0.0001$. Source data are provided as a Source Data file.

such as CAR-T cells and NK cells, via increase of granzyme secretion. G9a/GLP inhibition resulted in specific changes at the genetic and protein levels, and further research is required to fully elucidate the function of G9a/GLP in cytotoxic cell types. Overall, our results add to a growing body of evidence that supports epigenetic modulation of ACT products to improve antitumor efficiency, which in turn, could reduce the risk of adverse side-effects and the cost of ACT.

## Methods

### Ethics approval

All experimental procedures were approved by the Institutional Animal Care and Use Committee (IACUC) of A*STAR (Biopolis, Singapore) (IACUC No. 18139) in accordance with the guidelines of the Agri-Food & Veterinary Authority (AVA) and the National Advisory Committee for Laboratory Animal Research (NACLAR). NOD-SCID-IL2RGnull (NSG) mice (aged 8–10 weeks) were used for in vivo experiments, with 5 to 10 mice per group to ensure statistical power (further details in Supplementary Methods). Mice were euthanized humanely at defined study endpoints, as detailed in the IRB.

T cells isolated from patients diagnosed with HBV-related HCC HCC were obtained according to standards set by the Institutional Review Board (IRB: NUS-IRB H17-023E) and participants gave written informed consent, according to CARE guidelines and in compliance with the Declaration of Helsinki principles. Details of patient history can be found in our previous study[32].

### Study design

The objective of this study was to evaluate the potential therapeutic effect of small molecule inhibitors on T cells during ex vivo expansion. A panel of small molecule inhibitors was screened using the CellTox cytotoxicity assay (further details in below). After selection of the G9a/GLP inhibitor UNC0642 using different 2D and 3D cytotoxicity assays,

a range of in vitro *and* in vivo assays were performed to confirm the efficacy of UNC0642 in increasing the cytotoxicity of T cells against cancer cells (further details below). For in vitro studies, at least three independent experiments were carried out, and in the case of primary cells isolated from blood, at least three donors were tested unless otherwise indicated.

## Cell lines

HepG2-PreS1-GFP, HepG2.2.15, HepG2.2.15-luciferase, and Hep3b-GFP (from Prof. Antonio Bertoletti) cell lines were used in this study. Briefly, HepG2-PreS1-GFP cells were formed by transducing the HepG2 HCC cell line with a construct containing the preS1 portion of the genotype D HBV envelope protein gene covalently linked to GFP. HepG2.2.15 cell line supports the full HBV replication. All cells were cultured in 37°C and 5% $CO_2$ humidified incubator. Hepatitis B virus (HBV) surface antigen-producing HepG2-PreS1-GFP cells were cultured in R10 medium consisting of Roswell Park Memorial Institute (RPMI) 1640 medium without glutamine (Gibco, USA) supplemented with 10% heat-inactivated fetal bovine serum (FBS) (Gibco), 20 mM HEPES (Gibco), 100 U/mL penicillin, 100 µg/mL streptomycin (Gibco), 1× MEM amino acids solution (Gibco), 1 mM sodium pyruvate (Gibco), 1× MEM non-essential amino acids solution (Gibco), 1× GlutaMAX (Gibco), and 5 µg/mL plasmocin (Invitrogen, USA). Puromycin (5 µg/mL, TaKaRa, Japan) was added to the culture medium for cell selection. HBV-producing HepG2.2.15 and HepG2.2.15-luciferase cells were maintained in high glucose Dulbecco's modified Eagle's medium (DMEM) without glutamine (Gibco) supplemented with 10% heat-inactivated FBS, 100 U/mL penicillin, 100 µg/mL streptomycin, 1 mM sodium pyruvate, and 1× MEM non-essential amino acids solution. Geneticin (200 µg/mL, Gibco) was added to the culture medium for cell selection. Hep3b-GFP cells were cultured in DMEM without glutamine supplemented with 10% heat-inactivated FBS, 50 U/mL penicillin, 50 U/mL streptomycin, 1× GlutaMAX, and 1× mM sodium pyruvate.

## Effector cell isolation

Blood samples were obtained from healthy donors or patients who had undergone treatment for HBV-related HCC. Peripheral blood mononuclear cells (PBMCs) were isolated by Ficoll (GE Healthcare, USA) density gradient centrifugation according to the manufacturer's instructions and frozen down. In all, $1 \times 10^7$ PBMCs resuspended in 6 mL of AIM-V (Gibco) supplemented with 2% human AB serum (Sigma–Aldrich) were stimulated with 600 IU/mL IL-2 (Miltenyi Biotec) and 50 ng/mL anti-CD3 (eBioscience, USA) in each well of a 6 well plate for 9 days to allow T cell expansion at 37°C and 5% $CO_2$. The T cells were collected by centrifugation and then stored in liquid nitrogen prior to use in experiments. For NK cell isolation, PBMCs were thawed, and NK cells were isolated by negative selection using a human NK Cell Isolation Kit (130-092-657, Miltenyi Biotec, Germany) according to manufacturer's protocol. NK cells were then cultured in RPMI 1640 (Gibco) supplemented with 10% heat-inactivated FBS, 100 U/mL penicillin, and 100 µg/mL streptomycin.

## Epigenetic drug screening and treatment

A library of 24 epigenetic drugs was obtained from the Structural Genomics Consortium (SGC) Open Chemistry Networks platform (Open Chem Networks) was reconstituted in 100% DMSO (Sigma–Aldrich) to a stock concentration of 10 mM, then used at concentrations recommended by the consortium. $2 \times 10^6$ T cells per milliliter were cultured for 5 days at 37 °C and 5% $CO_2$ in the presence of drugs at concentrations ranging from 1 to 10 µM. The medium was refreshed on day 3 of the drug treatment. At the end of drug treatment, T cell-mediated target cell death was assessed using the CellTox assay. Data were normalized relative to untreated samples (0%) and lysis controls (100% cytotoxicity). Experiments were carried out with technical triplicates using three different donor T cells. Working stock

solutions of A366 (20 µM), UNC0638 and UNC0642 (both 5 µM) were prepared and serially diluted to the desired concentration with AIM-V. T cells were cultured in T cell expansion medium [AIM-V supplemented with 2% human AB serum and 600 IU/mL recombinant human (rh)IL-2] with the appropriate drug concentration for five days. T cells expanded 2–10 times the starting cell number, and varied among donors. Experiments with effector cells were conducted post-treatment with 1.25 µM of A366, UNC0638, and UNC0642.

## HBV-S183 antigen-specific T cell generation

HBV envelope S183–191 TCR mRNA was produced by in vitro transcription as previously described. $CD3^+$ T cells were cultured in the presence of 1000 IU/mL rhIL-2 for at least 6 h before electroporation using either 4D-Nucleofector (Lonza) or AgilePulse MAX (BTX). $CD4^+$ and $CD8^+$ T cells were not separated. For electroporation with 4D-Nucleofector, $1 \times 10^7$ T cells were suspended in 100 µL P3 Primary Cell Nucleofector Solution (Lonza) prepared according to manufacturer's instructions, whereas for electroporation with AgilePulseMAX, $1 \times 10^7$ T cells were suspended in 200 µL BTXpress Cytoporation Low Conductivity Medium T (BTX). HBV envelope S183–191 TCR mRNA (20 µg) was added per $1 \times 10^7$ of T cells and electroporated. After which, cells were allowed to recover overnight at 37 °C under 5% $CO_2$ in AIM-V supplemented with 10% human AB serum and 100 IU/mL rhIL-2. T cells expressing the introduced TCR were quantified by flow cytometry using either TCR Vβ3-FITC (Beckman Coulter, USA) or HLA-A201–HBV envelope 183–191–PE dextramer (Immudex) as described below. The S183-191 TCR is subsequently expressed on 20–80% of the $CD3^+$ cells, depending on the donor. Experiments were conducted only if at least 50% of $CD3^+$ cells expressed the S183-191 TCR.

## CAR-T cell generation

scFv-CD28-CD3ζ-T2A-RFP-anti-CD133 plasmid (Creative Biolabs, USA) was used to generate CD133-directed CAR-T cells from PBMCs. As a non-targeting CAR-T cell control, we generated CD19-directed CAR-T cells using an anti-CD19-scFv-4-1BB-CD3z-IRES-mCherry plasmid, kindly gifted from Prof. Michael Birnbaum's lab at SMART CAMP. Briefly, PBMCs were stimulated with 600 IU/mL IL-2 and 50 ng/mL anti-CD3 in AIM-V supplemented with 2% human AB serum for 4 days. CD133 or CD19 CAR-T lentivirus particles were added to the T cells at a multiplicity of infection (MOI) of 5 and cultured overnight in the presence of polybrene (8µg/mL) and Dynabeads human T-Activator CD3/28 according to manufacturer's protocol. T cells were then incubated for 4 days and T cells expressing RFP (CD133 CAR-T) or mCherry (CD19 CAR-T) were isolated by sorting with a BD FACSAria II (BD BioSciences) prior to expansion for 9 days in T cell expansion medium.

## Histone extraction and immunoblotting

Histones were extracted using the Abcam histone extraction protocol. In brief, cells were resuspended in Triton extraction buffer (0.5% Triton X-100 and 0.02% sodium azide in phosphate-buffered saline (PBS), supplemented with protease inhibitor) and lysed by incubation on a rocker at 4 °C for 10 min. Nuclei was pelleted by centrifugation at 6,500 g for 10 min in cold and washed with half the volume of Triton extraction buffer used for lysing. Acid extraction was then performed to isolate histones by resuspending nuclei pellet in 0.2 N hydrochloric acid and rocked overnight at 4 °C. The following day, cell debris were removed by centrifugation at $6500 \times g$ for 10 min in cold and supernatant containing the histones were neutralized with 2 M sodium hydroxide using one-tenth the volume of hydrochloric acid. Concentration was estimated using Bradford assay and 10 µg per sample loaded onto a 18% polyacrylamide gel for gel electrophoresis and transfer to a PVDF membrane using the Bio-Rad Mini-Protean system. Membranes were blocked with 5% blotting-grade blocker (#1706404, Bio-Rad) dissolved in 0.1% Tween-20 in PBS. (0.1% PBS-T) for 1 h, then probed with primary antibody overnight at 4°C. Membranes were then

washed with 0.1% PBS-T thrice for 5 min each, then incubated with the corresponding secondary antibodies for 1 h at room temperature. Membranes were washed again as above and treated with Pierce ECL Western Blotting Substrate (32106, Thermo Fisher Scientific) for 2 min, before being exposed to a medical X-ray film (Super RX-N, Fuji) and developed using a film developer. Antibodies used for immunoblotting are anti-H3K9me2 (31-1059-00, RevMAb), anti-H3K27me3 (07-449, Sigma-Aldrich) and anti-H3 (ab1791, Abcam). Quantification of bands was done using ImageJ software.

## Flow cytometric analysis

A minimum of $1 \times 10^5$ T cells were used for flow cytometry analysis. Various flow cytometry panels were used for different parts of the experiments (list of antibodies and dilutions provided in Supplementary Information). Live and dead cells were distinguished using LIVE/DEAD Fixable Near-IR Dead Cell Stain Kit (L10119, Invitrogen), followed by cell surface staining, and fixation with 1% formaldehyde (MP Biomedicals). If intracellular cytokine staining was required, cells were fixed and permeabilized simultaneously with BD Cytofix/Cytoperm (554722, BD Biosciences, USA), and intracellular cytokines were stained before fixation with 1% formaldehyde.

To detect transfected HBV envelope S183–191 TCR expression in vitro, cells were stained for the surface markers CD8 (BD Biosciences) and TCR Vβ3 (Beckman Coulter, USA) and then fixed. Flow cytometric data were then acquired on CytoFLEX (Beckman Coulter) or BD LSRII Analyzer (BD BioSciences). For in vivo experiments, cells were more stringently analyzed for the expression of the surface markers CD4, CD8 and HLA-A201–HBV envelope 183–91–PE dextramer (Immudex, Denmark).

To characterize the cytokine and exhaustion marker profiles of the T cells after drug treatment by flow cytometric analysis, T cells were either stimulated non-specifically by incubation for 5 h in the presence of Dynabeads CD3/28 (Gibco) or left unstimulated for the same time-period. For analysis of the cytokine profile, 3 µg/mL brefeldin A (eBioscience) was added 3–5 h prior to cell staining.

To induce S183 antigen-specific stimulation for the co-culture experiment, $1 \times 10^5$ HepG2-PreS1-GFP cells were seeded into a 96-well flat-bottomed plate and $1 \times 10^5$ S183-specific CD8+ TCR+ T cells were added. Brefeldin A was added 5 h before the T cells were harvested at the 24-h and 48-h time-points for flow cytometric analysis of the cytokine and exhaustion profile. All flow cytometry data analysis was performed with FlowJo software. Antibodies used and their corresponding dilutions are listed in Supplementary Table 1.

## 2D CellTox cytotoxicity assay

Cytotoxic effects were analyzed using 2D CellTox Green Cytotoxicity Assays (G8742, Promega) according to the manufacturer's protocol. Briefly, 5000 HBV-producing HepG2.2.15 hepatoma cells were seeded into a 96-well black/clear flat-bottomed tissue culture-treated imaging microplate (Falcon) and cultured overnight at 37 °C under 5% $CO_2$. 5000 drug-treated or untreated HBV-S183 antigen-specific T cells (CD8+ TCR+) were added to the microplate with appropriate controls. After adding the 2× CellTox Green dye diluted in assay buffer, the cells were incubated for 15 min before the fluorescence intensity of each well was measured at the various time-points using Biotek Synergy HT microplate reader.

## 2D xCELLigence assay

To test against S183-191 TCR-expressing T cells or CD133 CAR-T cells, $1 \times 10^5$ HepG2-PreS1-GFP or $5 \times 10^4$ Hep3b-GFP cells respectively were seeded into each well of an E-Plate VIEW16 after the baseline reading was recorded and inserted into the xCELLigence RTCA DP instrument (ACEA Biosciences). After incubation overnight at 37 °C under 5% $CO_2$, drug-treated or untreated effector cells were added at various effector-to-target ratios, and target cell lysis was monitored in real time for at

least 72 h. Impedance, which correlates with the adherence of the target cells to gold microelectrodes at the bottom of the plate, was measured every 15 min. The impedance reading was then converted into cell index by xCELLigence software. All experimental data were normalized to the cell index at the point of effector cell addition before analysis.

## 3D microfluidic device assay

Collagen hydrogels with $2 \times 10^6$ HepG2-PreS1-GFP or Hep3B-GFP cells/mL were prepared as previously described[23] and injected into dedicated 3D regions of a commercially available microfluidic AIM Chip (AIM Biotech). The hydrogel was polymerized by thermal cross-linking at 37 °C for 30 min. Subsequently, media channels in the hydrogel were hydrated with R10 medium and incubated for 24 h to permit interaction between HepG2-PreS1-GFP and the hydrogel microenvironment. The medium in the microfluidic device was then replaced with AIM-V medium supplemented with 2% human AB serum, 100 IU/mL rhIL-2, and 3 µM DRAQ7 (Miltenyi Biotec) in preparation for the first confocal imaging. $9 \times 10^4$ Electroporated HBV S183 antigen-redirected T cells or $6 \times 10^4$ NK cells labeled with 3 µM CellTracker Violet BMQC (Invitrogen) were added to one channel of the device and incubated for 24 h before the next confocal image acquisition. For devices containing Hep3B-GFP cells, $9 \times 10^4$ CD133 CAR-T cells were added per device without the need for labeling as they are also expressing mCherry.

## Confocal imaging and data analysis

All 3D killing assays were performed using an Opera Phenix High-Content Screening System by imaging constant hydrogel regions on Day 1 before the addition, and 24 h after the addition of electroporated T cells. The images were analyzed using Imaris Cell Imaging Software (Bitplane) by quantifying the number of live targets before, and 24 h after T cell addition. Live HepG2-PreS1-GFP were identified by GFP expression (green), dead cells were identified by DRAQ7 staining (red), T cells were labeled with BMQC (blue). The number of target cells killed or inhibited was quantified by subtraction of the number of baseline live target cells before the addition of T cells from the number of live target cells 24 h after the addition of T cells.

## Metabolic assays

Seahorse XFe96 Cell Mito Stress and Glycolysis Stress Tests (Seahorse, Agilent Technologies, Santa Clara, CA) were performed according to the manufacturer's instruction using a SeaHorse XF Extracellular Flux Analyzer (Agilent). After 5 days of drug treatment, T cells were washed twice with PBS, and resuspended in either Mito Stress seahorse XF RPMI assay medium (Seahorse Bioscience) supplemented with 10 mM glucose, 2 mM L-glutamine, 1 mM sodium pyruvate, or in Glycolytic stress seahorse XF RPMI assay medium supplemented with added 2 mM L-glutamine only. The cells were then seeded on a sterile XFp plate in quadruplicate ($5 \times 10^5$ cells/well) and starved for 1 h at 37 °C without $CO_2$ before starting the stress test. For the Mito Stress assay, cells were serially stimulated in medium containing the following reagents: (A) 5 µM oligomycin; (B) 8 µM carbonyl cyanide-4-(trifluoromethoxy) phenylhydrazone (FCCP); (C) 1 µM rotenone/antimycin A. For the glycolytic stress assay, cells were stimulated with: (A) 10 mM glucose; (B) 8 µM oligomycin; and (C) 50 mM 2-deoxy-D-glucose (2DG). For each time-point, the following respiratory parameters were recorded in pmol per minute: basal OCR, extracellular acidification rate (ECAR), ATP production, maximal respiratory capacity, spare respiratory capacity, glycolysis glycolytic capacity, and glycolytic reserve. Data [mean value ± standard error of the mean (SEM)] were analyzed using the Seahorse Wave software.

## Proteomics analysis

**Sample preparation.** T cells were prepared and treated as described above and then plated in different wells for collection every 24 h up to day 5. At each time-point, the samples collected were washed with

serum free medium and resuspended in denaturing lysis buffer (8 M urea, 50 mM HEPES) and stored at −80 °C prior to proteomic analysis, when samples were thawed, centrifuged and supernatants collected. Proteins were reduced in 50 µM Tris(2-carboxyethyl)phosphine hydrochloride (TCEP) for 20 min followed by alkylation in 50 mM chloroacetamide (CAA) for 20 min at room temperature. Subsequently, proteins were digested with 2.5 µg endoproteinase Lys-C for 4 h followed by the addition of 2.5 µg trypsin. The samples were then incubated overnight at room temperature on a shaker. Following digestion, the peptides were desalted using a HLB C18 resin column and vacuum-dried. Peptides were then reconstituted in 0.5 M triethylammonium bicarbonate buffer, multiplexed using a TMT kit (Thermo, MA, USA), according to the manufacturer's protocol and quenched with 10 mM ammonium formate. These peptides were then desalted, fractionated (14% to 60% in 10 mM ammonium formate) and washed twice in 0.1% formic acid, 65% acetonitrile (ACN) before mass spectrometry (MS) analysis.

**Mass spectrometry analysis.** Analysis was performed using Easy nLC1000 (Thermo) chromatography system coupled with an Q Exactive HF-X Quadrupole-Orbitrap MS system (Thermo). Each fraction was separated in a 70-minutes gradient elution (0.1% formic acid in water and 99.9% acetonitrile with 0.1% formic acid) using 50 cm × 75 µm ID Easy-Spray column (C18, 2 µm particles, Thermo). The following acquisition parameters were applied: data-dependent acquisition in positive mode with survey scan at 60,000 resolution, scan range of 350–1550 m/z, and automatic gain control (AGC) target of 3e6; collision energy of 36, MS/MS 45,000 resolution and AGC target of 1e5; isolation window at 1.0 m/z, and fixed first mass of 110 m/z.

**Data processing.** Peak lists for subsequent searches were generated in Proteome Discoverer 2.3 (Thermo Scientific) using Mascot 2.6.1 (Matrix Science) and concatenated with the forward/decoy Human Uniprot database. The following search parameters were applied: MS precursor mass tolerance of 20 ppm, MS/MS fragment mass tolerance of 0.04 Da, three missed cleavages; static modifications: carboamidomethyl (C); variable modifications: oxidation (M), deamidated (NQ), acetyl N-terminal protein, TMT6plex(N-term), TMT6plex(K). False discovery rates (FDR) were estimated at strict (1%) and medium (5%) levels.

**Data analysis.** The output data (protein abundance) from Proteome Discoverer were imported into the open-source R environment for data analysis and normalization. The day 0 (D0) condition was presented as a common condition in all of the multiplexes. Protein abundances were converted to Log2 abundances and normalized by first calculating a scaling factor per plex according to the medians of the common D0 conditions. The D0 conditions were averaged as per their replicates, and all conditions were normalized according to the median of the median abundances per condition. Only proteins with complete quantified abundance from D0 and at least one other paired time-point (control and treatment) (at least 9 quantified abundances) were retained for further analysis. Protein abundance from D0 to D5 was grouped by k-means clustering using the k-means function within the stats package in R. Clusters were determined as the total within squared distances.

**NanoString analysis**

**Sample preparation.** Post-treatment T cells were either stimulated by incubating for 5 h in the presence of CD3/28 Dynabeads (Gibco) to induce non-specific stimulation or left unstimulated through the time-period. The T cells were then washed with PBS and resuspended in 10 µL RLT buffer + β-mercaptoethanol. RNA from the lysed cells was amplified using CT1000 Touch Thermal Cycler (Bio-Rad) and hybridized with the default probes in the nCounter CAR-T Characterization Panel. A fully automated nCounter Prep station liquid-handling robot

was used to process the hybridized sample into the nCounter cartridge. The nCounter cartridge was then placed into the nCounter digital analyzer for direct digital counting according to the manufacturer's protocol. The counts were then exported and analyzed.

**NanoString analysis.** Differential expression and pathway scores were analyzed using nCounter Advanced Analysis Software (version 2.0.134). Differential expression was plotted as a volcano plot, and pathway scores were normalized to untreated samples before plotting.

**NanoString validation.** Increased *CCL1*, *CCL23* and *CCL18* gene expression was confirmed by qPCR (CCL1F: CGGAAGATGTGGACAGC AAG, CCL1R: GGCCTCTTTGCCTCTCTTCA; CCL23F: CATCTCCTACAC CCCACGAAG, CCL23R: GGGGTTGGCACAGAAACGTC; CCL18F: GGCCA GGAGTTGTGAGTT and CCL18R: GGTATAGACGAGGCAGCAGA) of T cells from 5 donors. The increases in CCL18 secreted by T cells from three donors were confirmed by ELISA following manufacturer's protocol (Human CCL18/PARC Quantikine ELISA Kit, R&D Systems, USA). In all, $1 \times 10^7$ UNC0642-treated or control primary T cells were fixed in 1% formalin at room temperature for 10 min on rocking followed by quenching in 125 mM glycine for 5 min. Cells were then pelleted and washed with ice-cold PBS. Cell lysis and DNA fragmentation was then performed by resuspending cells in 300 µL of SDS lysis buffer (50 mM Tris pH 8.0, 10 mM EDTA, 1% SDS) and sonicating on a Bioruptor (UCD-200, Diagenode) at 30 s ON/OFF cycles for 30–36 cycles. Cell debris was removed pelleting at 14,000 rpm for 10 min at 4 °C. For immuno-precipitation, 2 µg of anti-H3K9me2 (ab1220, Abcam) or anti-total H3 (ab1791, Abcam) was incubated with 10 µg of pre-cleared chromatin together with 20 ng of spike-in chromatin (53083, ActiveMotif) and 1 µg of spike-in antibody (61686, ActiveMotif) overnight on rotation in the cold. The next day, antibody was captured with 30 µL of Protein A beads for 4 h. Beads were then washed, eluted and decrossed. DNA was recovered using the ChIP DNA Clean & Concentrator kit (D5205, Zymo Research). Eluates were subjected to quantitative real-time PCR (qPCR) using the Maxima SYBR Green Master Mix (K0221, Thermo Fisher Scientific) on the CFX96 Real-PCR Detection System (Bio-Rad). hsCCL23 promoter – 5′-CTTCTCGGGATGCCAGTTCT-3′ (forward) and 5′-CCCAGAGGGAAGTATCCATCA-3′ (reverse) and hsCCL18 promoter – 5′-TGGTTCATTCCATTGGATGGT-3′ (forward) and 5′-AGACGCTGACCT TCGCACTT-3′ (reverse).

**In vivo xenograft tumor model and adoptive cell transfer.** NSG mice were purchased from InVivos Pte Ltd. and maintained under specific pathogen-free conditions. All animal experiments were approved by the Institutional Animal Care and Use Committee (IACUC) of A*STAR (Biopolis, Singapore) (IACUC No. 18139) in accordance with the guidelines of the Agri-Food & Veterinary Authority (AVA) and the National Advisory Committee for Laboratory Animal Research (NACLAR).

NSG mice (aged 8–10 weeks) were inoculated with HepG2.2.15 luciferase-expressing cells ($2 \times 10^6$ cells/mouse) by intrahepatic injection and the tumor was allowed to grow for 7 days. Before the adoptive cell transfer, the tumor was scanned by in vivo imaging (IVIS; Xenogen, Alameda, CA). Mice were then anesthetized, injected subcutaneously with 200 µL d-luciferin potassium salt (Caliper Life Sciences) (5 mg/mL in PBS) and, 2 min later, bioluminescence images were acquired. Bioluminescence in the tumor was quantified using Living Imaging 3.0 software. Based on luminescence, the mice were randomly distributed into the different groups (7 were designated for injection with drug-treated HBV S183 antigen-redirected T cells, 8 for drug-treated mock-electroporated T cells, 7 for untreated HBV S183 antigen-redirected T cells, 7 for untreated mock-electroporated T cells, and 3 without T cell injection). Untreated and UNC0642-treated HBV S183 antigen-redirected T cells had their cell numbers adjusted to $1 \times 10^6$ CD8+ TCR+ T cells and were resuspended in 100 µL PBS per mouse and

injected intravenously via the tail vein. The cell number of the treated and untreated mock-electroporated T cells were adjusted to match those of their S183 antigen-redirected counterpart. Tumor growth progression was monitored by IVIS every 2–3 days and T cells were injected every 2–3 days.

**Statistical analysis.** All statistical analyses were performed with Prism software version 8.0 (GraphPad). Statistical significance was determined using two-sided Mann–Whitney test (comparison of two datasets) or two-sided one-way analysis of variance (ANOVA) (comparison of more than two datasets) unless stated otherwise. Data obtained in experiments involving multiple donors were analyzed using paired or matched tests as described in the figure legends. All data is represented as the mean ± SEM unless indicated otherwise in the figure legends.

### Reporting summary
Further information on research design is available in the Nature Portfolio Reporting Summary linked to this article.

## Data availability
Source data are provided as a Source Data file as described in figure legends. Mass spectrometry data have been deposited to the jPOST repository[54] under the accession numbers JPST001995. These data are also available via ProteomeXchange under accession codes PXD039542. All other datasets generated during and/or analyzed during the current study are available in the article and its Supplementary files or from the corresponding author on request. Source data are provided with this paper.

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

## Acknowledgements

We want to thank Alrina Shin Min Tan for her help with the Seahorse experiments and Winnie Teo for supporting the mRNA production. Also, we want to thank Ginny Xiaohe Li for the support with the proteomic analysis and thanks to Christine Tham for her support with the Nanostring experiments. We want to thank Jyothsna Vasudevan for her help with figure illustrations. This work was supported by the Open Fund - Young Individual Research Grant (OF-YIRG) - OFYIRG18nov-0002 awarded to AP and GA; the Competitive research programme (CRP) by the National Research Foundation Singapore (NRF) NRF-CRP17-2017-06 awarded to A.P., E.G., A.B., and Q.C.; Singapore National Research Foundation, NRF-SIS "SingMass" grant to RMS awarded to R.S.; and by the Institute of Molecular and Cell Biology, Agency for Science Technology and Research (A*STAR), Singapore.

## Author contributions

A.P., G.A., M.S.Y.L., J.A.R.C., and J.R.O. designed the experiments. M.S.Y.L., J.A.R.C., J.R.O., J.J.Y.A., R.V., and D.T. performed the experiments, E.C., T.T., A.T.T., Q.C., F.L., and G.A. helped with some of the experimental procedures. Y.T.L., W.L.C., and R.M.S. performed the proteomic analysis. A.B., E.G. and A.P. were involved in conceptualize the work and funding acquisition. All authors participated in the writing of the manuscript and approved the final version.

## Competing interests

A.P. is a consultant and shareholder of AIM Biotech Pte. Ltd.; E.G. and T.T. are cofounders and scientific advisors of IMMUNOA Pte Ltd. T.T. is in the Board of Directors of IMMUNOA Pte Ltd; A.B. is cofounder of Lion TCR Pte Ltd, in the Board of Directors Lion TCR Pte Ltd and in the Board of Directors of IMMUNOA Pte Ltd; E.G. has served on advisory board for Lion TCR Pte Ltd. The remaining authors declare no competing interests.
