## [Peer Review File · Nature Communications]

G9a/GLP inhibition during ex vivo lymphocyte expansion increases in vivo cytotoxicity of engineered TCR-T cells against hepatocellular carcinomaREVIEWER COMMENTS

Reviewer #1 (Remarks to the Author):

In this manuscript the authors report the potential use of G9a/GLP inhibitors to enhance the activity of transient engineered TCR+ T cells. After screening several small molecules regulating epigenetic players, they focus on those inhibiting G9a/GLP. They further select among the possible inhibitors and using a condition in which cytotoxicity is increased without major effects on the T cell expansion or transient expression of TCR. They also evaluate the changes affecting different T cell populations by flow cytometry, targeted transcriptomics and proteomics. Finally they also show effects in orthotopic in vivo model of HCC. The authors also provide evidences suggesting that such G9a/GLP inhibition may also enhance the activity of CAR T cells (targeting CD133) and cytotoxic activity of NK cells. The work is interesting, and can be related to other reports indicating that G9a/GLP could be an attractive target for HCC and cholangiocarcinoma therapy. However it also show some discordances that require further clarification and some experimental data.

The analysis of the compounds on histone modifications should include not only on H3k9me1/2 but also H3K27 me1, me2 y me3 according to some reported (PMID 33436557, 31270502, 25168426, 25365549, 24389103) cross activities between G9a/GLP and PRC2. Authors should clarify whether Data in Fig S1A are obtained with different donor cells or a single one repeatedly treated

TCR+ T cell cytotoxicity after treatment with epigenetic drugs was evaluated using a microfluidic device, other approaches should also be used and, if possible, analyze cytotoxicity using different effector to target cell ratios. In particular in experiments such as those shown in Fig 1D

Of the epigenetic inhibitors, the authors observed that UNC0638 and A366 failed to increase cuiytotoxicity in some donors. While this lead the authors to focus on UNC0642, this observation may have important implication in the clinic, do the authors have any explanation? Is this related to the lack of response observed in T cells from one of the HBV-related HCC patient? This aspect deserves detailed discussion

The authors observed that UNC0642 treatment increases T cell, without modifying T cell subpopulations or T cell activation or exhaustion. However, Nanostring analyses revealed upregulation of CTLA4 and FoxP3, which appear to be discordant with flow cytometry analyses. This discrepancy should be clarified. Are different translational effects such as those observed for CCL18? How are those explained? Are other changes, such as those of GZMB, GZMM and perforin expression, observed in the proteomic study in correlation with the flow cytometry and targeted transcriptional profiling? Finally, a major caveat for the in vivo experiment is the use of of highly immunodeficient mice (NSG) where the activity of the treated TCR+ T cells in relation with other immune cell populations can not be studied in depth. A potential immunocompetent mouse model is hagly recommended, Alternatively, did the authors tried to establish such orthotopic models in a less restricted immunodeficient models, such as nude mice? In this, only T cells are lacking and possible crosstalk can be evaluated.

Reviewer #2 (Remarks to the Author):

The authors performed a screening of small molecules that regulate epigenetic cell control for increased cytotoxicity and claim that low-dose inhibition by G9a/GLP by exposure to a compound denominated UNC0642 during T cell generation (ex vivo expansion) resulted in increase T cell cytotoxicity, likely through increased GZMB expression, in in vitro and in vivo assays. In its current form, the data presented does not specifically support the claim that that UNC0642 can increase the redirected anti-tumor efficiency of this approach.

Major comments:

1. In Figure 1B the authors show the data as a relative fluorescence intensity compared to untreated. It would be helpful for the readers to understand what is the actual data – how much is 100% and 0% - to have an idea of what absolute interval is actually shown. This comment also applies to other

figures where the same data manipulation is used.

2. Fig 1D, 1I, 1J, Fig 5C, Fig 6C, G, J: To show that the compound is having a specific effect on tumor killing mediated by the antigen-specific T cells, the authors should show an additional control which is non-electroporated (or mock-electroporated) T cells plus the compound. This would eliminate the doubt that the effect of UNC0642 is not specific and would be the same on non-transfected T cells.
3. Fig 5C: The authors rightly point out that there is no difference in tumor control in the mouse model between No T cells and TCR transfected T cells that are not treated with UNC0642 and offer the explanation that this could be due to the immune-tolerant environment of the liver. However, another interpretation is that the T cells are not effective in this model and UNC0642 acts non-specifically on non-transfected T cells. The authors should perform experiments to tease this possibility out.
4. The histologic evidence presented in Figure 5E is underwhelming (there are 2 lymphocyte clusters identified in each and could be due to a sampling effect). Could the authors perform a general quantification of T cell infiltration in the tumors?
5. Although the authors claim the differences in patient #2 are dramatic (Fig 6C, bottom panel), they seem minimal to me at least in the 2D assays. The data for the CAR T cell model are non-significant and it's unclear if treating with UNC0642 in the NK model has any effect. Again, the data would be better supported by a non-EP compound treated control.

Minor comments:

1. Figure 1A and the related text needs a better description of the T cell generation process. What is proportion of CD4 and CD8 T cells in the product? What is the level of TCR expression after TCR transfection? What is the fold expansion? What are the cells used? Also the authors use both CD8 and CD4 T cells – is the specific TCR they are using functional in CD4 T cells (ie do they induce CD4 T cells to kill tumors)? If so, the authors should show the data.
2. The authors should avoid mentioning commercial assays without describing in the text what is the underlying basis of the assay – for example – how does a “2D cell tox assay” function and what does it measure.
3. Fig 1B: Please clarify if the data is normalized to the % transduction of the cells or cell viability.
4. Supp Fig 1B: please clarify WB to the right and left of figure (before and after drug exposure?).
5. Line 103: please clarify what “ a better pharmacokinetic profile” signifies as this is critical info to understand choice of the compound.
6. Figure 1D: please define EP (electroporation)
7. Line 310: the authors identified 2 patients – please clarify their treatment history and also their CBC at the time of T cell extraction to confirm these were abnormal compared to normal donors.

Reviewer #3 (Remarks to the Author):

In the present study the authors aim to identify epigenetic regulators that may affect the antitumor efficacy of engineered T cells. By screening a small set of small molecules targeting epigenetic regulators using in vitro assays they demonstrated that treatment of G9a/GLP inhibitor UNC0642 increased T cell antitumor activity. They further showed that UNC treatment increased expression of GZMB and IL-2, but not PERP. This result is consistent with the finding with the NanoString nCounter CAR-T related gene expression panel showing that UNC changes chemokine expression and cytotoxicity pathways at the transcription level. Furthermore, the authors showed that UNC treatment also increased expression of proteins associated with T cell activation and T cell activation. Finally, by using both an orthotopic mouse model or T cells from patient donor they showed that UNC treatment improves CAR-T cell mediated tumor killing.

Overall, the experiments are well designed and findings are novel and clinically relevant. However, a few issues need to be addressed.

Major Concerns:

In Figure 1B, what is the rationale to choose the 24 molecule inhibitors? Any specific reason or focus? Also, it is unclear how many times this experiments were repeated and how reproducible the results were. This information needs to be indicated in the figure legends.

In Figure 2A, it is unclear how many TCR-engineered T cells were used in the analysis. Also, the FACS results of GzmB expression need to be validated by an independent approach such as RT-qPCR or WB.

In Figure 3C, it is unclear why UNC0642 treatment resulted in a two fold increase in the level of H3 in the CCL23 locus. Any explanation or these experiments should be repeated to make sure the results are reproducible.

In Figures 5A-D, the data are interesting. Given that G9a/GLP also impacts gene expression and growth of tumor cells and that multiple injection of UNC0642-treated TCR+ T cells, it is important to know whether or not UNC0642 in the injected T cells was removed before injection into the mice. This information is missing in the figure legends. This information is also missing in Figure 6.

Minor Concerns:

In the figure legend of Figure 3C, CCL28 should be CCL23.

Should CTLA4 be a T cell costimulatory molecule or immune checkpoint effector?

Reviewer #1

In this manuscript the authors report the potential use of G9a/GLP inhibitors to enhance the activity of transient engineered TCR+ T cells. After screening several small molecules regulating epigenetic players, they focus on those inhibiting G9a/GLP. They further select among the possible inhibitors and using a condition in which cytotoxicity is increased without major effects on the T cell expansion or transient expression of TCR. They also evaluate the changes affecting different T cell populations by flow cytometry, targeted transcriptomics and proteomics. Finally they also show effects in orthotopic in vivo model of HCC. The authors also provide evidences suggesting that such G9a/GLP inhibition may also enhance the activity of CAR T cells (targeting CD133) and cytotoxic activity of NK cells.

The work is interesting, and can be related to other reports indicating that G9a/GLP could be an attractive target for HCC and cholangiocarcinoma therapy. However it also show some discordances that require further clarification and some experimental data.

The analysis of the compounds on histone modifications should include not only on H3K9me1/2 but also H3K27 me1, me2 y me3 according to some reported (PMID 33436557, 31270502, 25168426, 25365549, 24389103) cross activities between G9a/GLP and PRC2.

Response:

To address the reviewer's comment we read the suggested literature and we carried out an additional experimental analysis of H3K27me3 after UNCO624 treatment. We selected H3K27me3 because most of the papers suggested by the reviewer considered H3K27me3 as the key indicator of PRC2 activity. As shown in the new data, highlighted in the revised version of the manuscript, we found no change

in H3K27me3 levels after treatment (Supp. Fig1 D reported aside for reference). This is consistent with the papers the reviewer has highlighted where G9a inhibition has minimal impact on H3K27 methylation as summarized here for clarity. PMID 33436557 showed minimal to no change in H3K27me1/2/3 in all the tested myeloma cells lines treated with UNCO638 alone (another G9a/GLP inhibitor); PMID 24389103 found that UNCO642 treatment of mouse embryonic stem cells induced global decrease in H3K9me2 but not in H3K27me3 levels; PMID 31270502 showed a decrease in H3K27me3 in bladder cancer cells using a G9a/DNMT dual inhibitor but found no change in hepatocarcinoma or acute lymphoblastic leukemia; PMID 25168426 did not study the effect of G9a but EZH2 on H3K27 methylation. PMID 25365549 is a review that highlights that the

relationship between G9a and PRC2 is complicated and cell type-specific, however, it does not discuss the role of G9a or PRC2 in immune cells, and more specifically cytotoxic T cells.

Although the analysis of H3K27me3 in this context aids in understanding how G9a is involved in histone modification in different cell types, we wish to highlight that our data may not be directly comparable to others in literature that thoroughly investigate G9a activity using total knock-out or full pharmacological inhibition in cancer cells or stem cells. In contrast, our study exploits a partial and short-term inhibition of G9a/GLP in transient cytotoxic T cells (TCR T cells). In fact, H3K9me2, which is the main read-out for G9a/GLP as reported in the suggested papers, was only reduced by 50% in our study (Supp Fig.1C reported aside for reference).

Authors should clarify whether Data in Fig S1A are obtained with different donor cells or a single one repeatedly treated.

Response:

We thank the reviewer for his/her comment that allows us to clarify this point. The experiments were carried out with different donor cells as we have now clarified in the revised figure legends.

TCR+ T cell cytotoxicity after treatment with epigenetic drugs was evaluated using a microfluidic device, other approaches should also be used and, if possible, analyze cytotoxicity using different effector to target cell ratios. In particular in experiments such as those shown in Fig 1D.

Response:

We agree with the reviewer on using different approaches and we have in fact evaluated engineered TCR+ T cell cytotoxicity using traditional 2D assays where cell death is indicated by a change in fluorescence (on the right, Figure 1B showing the CellTox assay, details in Supplementary Methods) or a change in cell impedance (below Figure 1D-G

showing the xCelligence assay, details in Supplementary Methods). In our revision, we have now analysed additional untreated and UNC0642-treated TCR- T cell controls (Figure 1D-G)

and used different effector: target cell ratios (Supp Figure 1H-L, below) as requested by the reviewer. Changes have been highlighted in the revised text.

Of note, our current and previous work has compared the use of 3D assays to 2D assays, and we have shown that 3D cytotoxicity assays are more predictive of *in vivo* T cell cytotoxicity behaviour, as they can recapitulate a physical environment for T cells to actively migrate to reach target cells (Pavesi et al, 2017. PMID: 28614795).

Of the epigenetic inhibitors, the authors observed that UNC0638 and A366 failed to increase cytotoxicity in some donors. While this led the authors to focus on UNC0642, this observation may have important implications in the clinic, do the authors have any explanation? Is this related to the lack of response observed in T cells from one of the HBV-related HCC patients? This aspect deserves detailed discussion

Response:

The comparison of UNC0638, A366 and UNC0642 was carried out in our 3D *in vitro* assay as the 3D environment is more challenging for T cells and separates T cell function better than traditional 2D assays (Pavesi et al, 2017. PMID: 28614795). Besides performing better in the 3D *in vitro* assays, UNC0642 has a better pharmacokinetic (PK) profile. Although we designed the protocol such that UNC0642 is removed from the engineered T cells before they are introduced into patient, our findings will be informative to other labs where it is not uncommon to co-inject drugs with engineered T cells to improve their efficacy. In such a case, a drug with a good PK value would be required.

We thank the reviewer for noticing the clinical implication of our observation. We hypothesize that the lack of response to UNC0642 treatment in one of the HBV-related HCC patients (Patient 1) is based on how well the patient's T cells performed after the TCR engineering. We observed that when the

engineered T cells were more effectively killing the target cells, the increase in cytotoxicity due to UNC0642 treatment was not dramatic (Revised Figure 6D-E, top panels). In contrast, in the other patient (Patient 2) whose T cells did not perform so well after TCR engineering, UNC0642 treatment significantly improved T cell cytotoxicity (Revised Figure 6D-E, bottom panels).

A similar trend was observed in our experiments with engineered TCR+ T cells at different effector:target ratios, where at higher effector:target ratios the effect of UNC0642 treatment was less dramatic (Figure 1D). While at lower effector:target ratios, UNC0642 treatment was more dramatic (Revised Supp Figure 1H-L).

We have now revised the main text to include these possible explanations.

The authors observed that UNC0642 treatment increases T cell, without modifying T cell subpopulations or T cell activation or exhaustion. However, Nanostring analyses revealed upregulation of CTLA4 and FoxP3, which appear to be discordant with flow cytometry analyses. This discrepancy should be clarified.

Response:

We did observe a small increase in CTLA4 expression by flow cytometry analysis in T cell CD4+ and CD8+ populations when analyzed separately (Figure 2E, on the left), matching the Nanostring analysis. The increase was not clear when both populations were pooled, likely due to the differences in the degree of change in each subpopulation. However, we did not assay for FoxP3 by flow cytometry as we selected CD25 as a marker for Treg cells (Supp Figure 2B, reported below), and found no significant change in the Treg population after UNC0642 treatment.

Based on previous evidence in literature, we consider the increase in *CTLA4* and *FoxP3* indicated in the Nanostring analysis as relatively small, considering that *CTLA4* and *FoxP3* can be expressed after T cell activation, where their increase at small levels are not indicative of a hyporesponsive state in T cells (*CTLA4* expression and exhaustion reviewed in PMID: 26167163 and PMID: 26205583; *FoxP3* expression and Treg function in PMID: 17154262, PMID: 17329235 and PMID: 19849846).

Notably, we observed a much more dramatic increase in granzyme B expression in the total CD3 population by flow cytometry and proteomic analyses (Figure 2B and 4A, reported below), which was also observed in CD4+ and CD8+ subpopulations (Figure 2D, below).

This increase was sustained after antigen stimulation by target cell co-culture (Figure 2F-G, reported below).

Given the complicated suite of markers that can be used to classify cell states, we chose to use proliferation and cytotoxicity as an indicator of cell state instead. We, therefore, concluded that UNC0642 did not affect cell states or proliferation, but increased cell granzyme expression and T cell cytotoxicity.

We have now increased the robustness of our data for CD4+ and CD8+ subtyping by repeating the profiling with additional donor T cells (revised Supp Figure 2B-D, below). No changes to the subtype populations were found confirming our previous data.

Are different translational effects such as those observed for CCL18? How are those explained?

Response:

The proteomics screen that was carried out is unable to pick up secreted cytokines such as CCL18, hence it was not a hit, despite validating that its expression (Fig 3B and C, reported below) and its secreted protein (Fig 3D and E, reported below) is increased after UNC0642 treatment.

Papers have described it as both promoting and antagonizing T cell effector behaviour, depending on the context. In our discussion, we highlight that CCL18 can undergo significant post-translational modification that alters its function, and that its function is highly context-dependent. Significant additional work is required to tease out the function of secreted CCL18 on T cell effector function well beyond the scope of the present study.

Are other changes, such as those of GZMB, GZMM and perforin expression, observed in the proteomic study in correlation with the flow cytometry and targeted transcriptional profiling?

Response:

Yes, genes involved in cytotoxicity were upregulated at the gene (Fig 3F) and protein level (Fig 4A and B), and were also validated by flow cytometry (Figure 2B, 2F-G and 4C). Together with the observed increase in T cell-mediated target cell death after UNC0642 treatment, without any change in T cell migration, we concluded that UNC0642 treatment increases T cell cytotoxicity by increasing effector cytotoxicity functions such as granzyme secretion.

Finally, a major caveat for the *in vivo* experiment is the use of highly immunodeficient mice (NSG) where the activity of the treated TCR+ T cells in relation with other immune cell populations cannot be studied in depth. A potential immunocompetent mouse model is highly recommended. Alternatively, did the authors try to establish such orthotopic models in a less restricted immunodeficient model, such as nude mice? In this, only T cells are lacking and possible crosstalk can be evaluated.

Response:

NOD-*scid* *Il2rg*^{null} (NSG) mice were utilized in generating an orthotopic liver cancer model for the validation of treated TCR + T cell therapy *in vivo* as they lack mouse immune components such as T, B, NK cells, and some other myeloid subsets that may potentially interfere with the functions of injected human T cells. We agree that the function of the treated TCR+ T cells could change due to a crosstalk with other immune cell populations and this could result in a synergistic anti-tumour effect. However, to evaluate the efficacy of our drug-treated engineered TCR⁺ T cells, we preferred a widely recognized immunodeficient animal model to understand the impact of the redirected T cells in targeting liver cancers without any “noise” caused by the host immune system. In the future, Transgenic, Wild-Type and/or even humanized mice could be considered to identify mechanisms of interaction with immune components and their potential role in suppressing/eliminating the tumor cells.

Reviewer #2 (Remarks to the Author):

The authors performed a screening of small molecules that regulate epigenetic cell control for increased cytotoxicity and claim that low-dose inhibition by G9a/GLP by exposure to a compound denominated UNC0642 during T cell generation (ex vivo expansion) resulted in increase T cell cytotoxicity, likely through increased GZMB expression, in in vitro and in vivo assays. In its current form, the data presented does not specifically support the claim that UNC0642 can increase the redirected anti-tumor efficiency of this approach.

Response:

We believe that with the revised *in vitro* and *in vivo* data including additional controls, the reviewer will agree that indeed anti-tumor efficiency increases with this approach. In this study, we presented multiple *in vitro* and *in vivo* cytotoxicity assays that show a clear improvement of T cell cytotoxicity after UNC0642 treatment. Specifically, target cell killing data with engineered TCR+ T cells are reported into the revised Figures 1D-G, 1I, 7C-D and Supp Figures 1H-L for *in vitro* results and revised Figure 5 for *in vivo* results; target cell killing by engineered CAR-T cells are reported into the revised Figure 7C-E; non-specific tumor cell killing by NK cells is reported into the revised Figure 8C-E.

We would like to clarify that we do in fact think that UNC0642 increases granzyme B expression and cytotoxicity generally in both non-transduced and transduced T cells populations. This should be clearer now with the revised experiments that include non-transduced T cells (addressed in detail below in our response to the major comments 2 and 3). Additionally, UNC0642 improved granzyme expression and non-targeted cytotoxicity in NK cells (Revised Figure 8B-E, reported below for reference). Hence, it is not a phenomenon that is restricted only to redirected engineered T cells.

Major comments:

1. In Figure 1B the authors show the data as a relative fluorescence intensity compared to untreated. It would be helpful for the readers to understand what is the actual data – how much is 100% and 0% - to have an idea of what absolute interval is actually shown. This comment also applies to other figures where the same data manipulation is used.

Response:

We choose a mathematical normalization of the data to show the observed differences. The data were expressed as relative fluorescence intensity considering the fluorescence intensity of the untreated sample as reference. It was stated in the graph that the data were normalized such that 0% is untreated control while 100% is the lysis control. However, to address reviewer comment, we have

now plotted the data as absolute intensities in the revised Figure 1B.

B

2. Fig 1D, 1I, 1J, Fig 5C, Fig 6C, G, J: To show that the compound is having a specific effect on tumor killing mediated by the antigen-specific T cells, the authors should show an additional control which is non-electroporated (or mock-electroporated) T cells plus the compound. This would eliminate the doubt that the effect of UNC0642 is not specific and would be the same on non-transfected T cells.

Response:

We have now repeated the cytotoxicity experiments to include this additional control as requested by the reviewer. UNC0642 treatment increased T cell cytotoxicity in mock-electroporated TCR- T cells, as well as electroporated TCR+ T cells. We have now added the results in Figure 1D-G, and included here below for the reviewer's reference.

D

E

F

G

Additionally, we previously reported that UNC0642 treatment increases granzyme B expression in T cells immediately after *ex vivo* expansion, before TCR transduction (Figure 2F-G, reported below), which is sustained after TCR transduction (Figure 2G). Granzyme B expression is a good indicator of T cell potential cytotoxicity, and the increased Granzyme B expression is consistent with the increase in functional target cell killing.

Overall the previous and revised data show that UNC0642 treatment increases T cell cytotoxicity generally, and, in the context of engineered T cells, it can further improve their targeted anti-tumor activity.

3. Fig 5C: The authors rightly point out that there is no difference in tumor control in the mouse model between No T cells and TCR transfected T cells that are not treated with UNC0642 and offer the explanation that this could be due to the immune-tolerant environment of the liver. However, another interpretation is that the T cells are not effective in this model and UNC0642 acts non-specifically on non-transfected T cells. The authors should perform experiments to tease this possibility out.

Response:

As requested by the reviewer, we have performed experiments in the mouse orthotopic model with the additional control of UNC0624-treated non-transduced (mock electroporated) T cells. Additionally, we have increased the number of mice to have better statistical power and the number of timepoints to understand the persistence of the treated TCR⁺ T cells. The additional results are presented in the revised Figure 5. With this improved experimental design, we observed that TCR⁺ T cells are better at restricting tumor growth than TCR⁻ T cells. Consistent with our revised *in vitro* data, UNC0642 did increase T cell cytotoxicity even in non-transfected T cells *in vivo*. However, treated non-transfected T cells were not able to restrict tumor growth to the same extent as treated transfected T cells. Notably, we observed a significant tumor growth at 22 days after intrahepatic injection of tumor cells in all conditions except for treated TCR⁺ T cells, suggesting that UNC0642-treated TCR⁺ T cells have an increased persistence of T cell cytotoxicity.

4. The histologic evidence presented in Figure 5E is underwhelming (there are 2 lymphocyte clusters identified in each and could be due to a sampling effect). Could the authors perform a general quantification of T cell infiltration in the tumors?

Response:

To address the reviewer comment, we have analyzed the CD3⁺ regions in multiple liver sections and the data is now added to the revised Figure 5E. The results show an increase in CD3⁺ regions in liver sections for mice with UNC0642-treated TCR⁺ T cells.

5. Although the authors claim the differences in patient #2 are dramatic (Fig 6C, bottom panel), they seem minimal to me at least in the 2D assays. The data for the CAR T cell model are non-significant and it's unclear if treating with UNC0642 in the NK model has any effect. Again, the data would be better supported by a non-EP compound treated control.

Response:

We regret that we are unable to repeat the experiment with additional non-EP compound-treated controls due to shortage of patient samples. We agree with the reviewer that the impact of UNC0642 treatment in 2D cytotoxicity assays are less dramatic. In our present and previous papers, we have highlighted that 2D cytotoxicity assays tend to overestimate the cytotoxicity of T-cell therapies, because a large part of effector-mediated killing is facilitated by gravity (PMID: 29559973; PMID: 28614795). 3D cytotoxicity assays are more informative as they impose additional obstacles and require T cells to actively migrate through an extracellular matrix to locate and kill the target cell, as they would do within an *in vivo* tumor. This is an important message included in our paper, and the motivation for showing the 2D and 3D cytotoxicity assays side-by-side.

Nonetheless, there is a clear difference between untreated and UNC0642-treated TCR+ T cells cytotoxicity in the 2D cytotoxicity assay (Figure 6D, bottom panels), which is even clearer in the 3D cytotoxicity assay (Figure 6E, bottom panels). Figure 6D-E reported below for reference.

As requested by the reviewer in minor point 7, we have now included the patients' CBC values (Figure 6B reported below) and modified the text accordingly, while the patients' treatment history has been published elsewhere (Tan et al, 2019. PMID: 30711630) and is now cited in the revised main text. The reviewer will note that Patient #1 had a lymphocyte count in the normal range and responded well to adoptive T cell therapy after TCR engineering. In contrast, Patient #2 had a low lymphocyte count and presumably compromised lymphocytes. As a likely consequence, we expected Patient #2 T cells to not perform well after TCR engineering. It is therefore remarkable that we observed a significant improvement in TCR-mediated target cell killing after UNC0642 for Patient #2.

	Reference Range	Patient 1	Patient 2
Monocyte Count (10 ⁹ /L)	0.2-0.8	0.7	0.6
Basophil Count (10 ⁹ /L)	0-0.1	0.11	0.02
Eosinophil Count (10 ⁹ /L)	0.04-0.44	0.24	0.06
Platelet Count (10 ⁹ /L)	140-440	300	160
Neutrophil Count (10 ⁹ /L)	2.0-7.5	3.82	4.47
Lymphocyte Count (10 ⁹ /L)	1.0-3.0	2.16	0.78
Platelet/Lymphocyte Ratio	124-199	138.89	205.13
Neutrophil/Lymphocyte Ratio	1.6-3.1	1.77	5.73
Lymphocyte/Monocyte Ratio	5.2-8.4	3.09	1.3

We have performed additional CAR-T experiments with multiple effector:target ratios and additional non-targeting CAR-T controls to improve the statistical power of the study (as reported in the revised Figure 7, Figure 7D-E included below for easy reference). The cytotoxicity data tends to be more convincing in the 3D assay as it is better at comparing T cell functionality for different experimental

conditions than 2D assays. In any case, the 2D and 3D data collected from these additional experiments both confirmed that UNC0642 treatment improves CAR-T cell cytotoxicity, and that UNC0642 treatment increases cytotoxicity even further in CD133-directed CAR-T cells.

We have also performed additional 2D cytotoxicity assays for the NK cell experiments to improve the statistical power of the study, with multiple effector:target ratios and demonstrated that UNC0642 treatment increases NK cell cytotoxicity in all tested ratios. Further, we have carried out 3D cytotoxicity assays for NK cells, even though NK cells were not engineered to target tumor cells specifically, and we observed increased cytotoxicity upon UNC0642 treatment. These results are reported in the revised Figure 8.

Minor comments:

- Figure 1A and the related text needs a better description of the T cell generation process. What is proportion of CD4+ and CD8+ T cells in the product? What is the level of TCR expression after TCR transfection? What is the fold expansion? What are the cells used? Also the authors use both CD8+ and CD4+ T cells – is the specific TCR they are using functional in CD4+ T cells (ie do they induce CD4+ T cells to kill tumors)? If so, the authors should show the data.

Response:

We apologize if the listed information on T cell generation was not explicit in the text. We have now included a brief description in the main text, and details in the Supplementary Methods. TCR⁺ T cells were produced according to the process described in Koh et al, 2013, referenced in the main text. Of note, the authors did an extensive testing of TCR function in CD4⁺ and CD8⁺ T cells, and found that both subtypes were functional. In particular, they observed that a mix of CD4⁺ and CD8⁺ TCR⁺ T cells was most effective in tumor killing. Following the previous results and to simulate clinical practice that does not segregate CD4⁺ and CD8⁺ populations before therapy, we did not separate both populations for our functional assays. The percentage of CD8⁺ cells of the total CD3⁺ T cell population assayed range from 20-90% (Supp Figure 2A, reported aside).

HepG2-PreS1-GFP cells were formed by transducing the HepG2 HCC cell line with a construct containing the preS1 portion of the genotype D HBV envelope protein gene covalently linked to GFP. T cells expanded 2 – 10 times from the starting cell number, and varied among donors (Supp Figure 1A, reported below, and Supplementary Methods). Depending on the donor, S183-191 TCR expression after electroporation ranges from 20 – 90% of the CD3⁺ cells (details now added in Supplementary Methods). However, experiments were only conducted if at least 50% of CD3⁺ cells expressed the S183-191 TCR (Supp Figure 1E).

We have now revised the Supplementary Methods to include the requested details on target cell line and T cell generation process.

- The authors should avoid mentioning commercial assays without describing in the text what is the underlying basis of the assay – for example – how does a “2D cell tox assay” function and what does it measure.

Response:

CellTox™ Green is a proprietary kit from Promega that measures cell death using a dye that only binds to cellular DNA and emits fluorescence when the cell membrane has been compromised. This allows us to identify conditions that lead to target cell death and differentiate it from low target cell numbers due to a cytostatic effect. We have now amended the main text to include a brief description and the main Methods section to refer to the Supplementary methods for further details on this assay.

- Fig 1B: Please clarify if the data is normalized to the % transduction of the cells or cell viability.

Response:

The data is normalized to the fluorescence signal in untreated control (0%) and the lysis control (100%). The data is not normalized to the % transduction of the cells as the same batch of transduced TCR⁺ T cells were used for all the wells. We have revised the figure to show the fluorescence intensities to avoid confusion.

4. Supp Fig 1B: please clarify WB to the right and left of figure (before and after drug exposure?).

Response:

Left image are control and UNC0642-treated samples, while right image are control, A366- and UNC0638-treated samples. We have revised the figure to make this clearer.

5. Line 103: please clarify what “ a better pharmacokinetic profile” signifies as this is critical info to understand choice of the compound.

Response:

We have revised the main text to clarify this point. Although we designed the protocol such that UNC0642 is removed from the engineered T cells before they are introduced into patient, our findings will be informative also to other labs that co-injected drugs with engineered T cells to improve their efficacy. In such a case, a drug with a good pharmacokinetic profile, such as UNC0642, would be preferred, as it indicates the compound has good bioavailability after being administered into the body.

6. Figure 1D: please define EP (electroporation)

Response:

We have revised the figure legends to include the shorthand definition.

7. Line 310: the authors identified 2 patients – please clarify their treatment history and also their CBC at the time of T cell extraction to confirm these were abnormal compared to normal donors.

Response:

Patients’ treatment history has been previously published elsewhere (Tan et al, 2019. PMID: 30711630) and has been referenced in the text. CBC data is now included in the revised Figure 6 and in the text as requested by the reviewer (See reply to the major comment 5).

Reviewer #3 (Remarks to the Author):

In the present study the authors aim to identify epigenetic regulators that may affect the antitumor efficacy of engineered T cells. By screening a small set of small molecules targeting epigenetic regulators using in vitro assays they demonstrated that treatment of G9a/GLP inhibitor UNC0642 increased T cell antitumor activity. They further showed that UNC treatment increased expression of GZMB and IL-2, but not PERF. This result is consistent with the finding with the NanoString nCounter CAR-T related gene expression panel showing that UNC changes chemokine expression and cytotoxicity pathways at the transcription level. Furthermore, the authors showed that UNC treatment also increased expression of proteins associated with T cell activation and T cell activation. Finally, by using both an orthotopic mouse model or T cells from patient donor they showed that UNC treatment improves CAR-T cell mediated tumor killing.

Overall, the experiments are well designed and findings are novel and clinically relevant. However, a few issues need to be addressed.

We thank the reviewer for his/her words of appreciation on our study and findings. We have absorbed the feedback and advice to improve the manuscript.

Major Concerns:

In Figure 1B, what is the rationale to choose the 24 molecule inhibitors? Any specific reason or focus? Also, it is unclear how many times this experiments were repeated and how reproducible the results were. This information needs to be indicated in the figure legends.

Response:

We thank the reviewer to allow us to clarify this point. The small molecule panel was obtained through the Structural Genomics Consortium (SGC) Open Chemistry Networks platform (Open Chem Networks). They are a public-private consortium that provides reagents targeting epigenetic and chromatin cofactors for the purpose of promoting open and innovative research. The panel was designed to target most key proteins involved in epigenetic regulation (readers-erasers-writers), and concentrations were used according to the consortium recommendation. The experiment was carried out with technical triplicates using three different donor T cells. We have edited the Supplementary Methods section and the corresponding figure and legend to reflect this information.

In Figure 2A, it is unclear how many TCR-engineered T cells were used in the analysis. Also, the FACS results of GzmB expression need to be validated by an independent approach such as RT-qPCR or WB.

Response:

For Figure 2A-E the data was acquired immediately after UNC0642 treatment, and the T cells used in the analysis were not transduced for TCR expression. For all the flow cytometry studies conducted in the manuscript, the minimum amount of cells analyzed is 100k. We validated the FACS results of GzmB expression after treatment by Mass Spectrometry (Figures 2F-G, reported below).

The proteomics data also showed that GzmB is the most upregulated protein appearing in the plot (Figure 4A and C; reported below). PCR and WB could result in a missed correlation with the protein expressed, therefore, we selected Mass Spectrometry as the best approach to validate our FACS results.

In Figure 3C, it is unclear why UNC0642 treatment resulted in a two fold increase in the level of H3 in the CCL23 locus. Any explanation or these experiments should be repeated to make sure the results are reproducible.

Response:

While the mean appears to be deviated from 1 (representing no change with the untreated sample), the difference is actually not statistically significant, once accounting for the variability in the mean as indicated by the error bars. Hence, we would claim that there is no change in H3 at CCL23. In contrast, the amount of H3K9me2 at CCL23 and CCL18 are decreased significantly from 1, with low variability in the mean, and therefore the change was statistically significant as indicated on the graph.

In Figures 5A-D, the data are interesting. Given that G9a/GLP also impacts gene expression and growth of tumor cells and that multiple injection of UNC0642-treated TCR+ T cells, it is important to know whether or not UNC0642 in the injected T cells was removed before injection into the mice. This information is missing in the figure legends. This information is also missing in Figure 6.

Response:

We apologize that this information was missing. To clarify, UNC0642 was added to the T cells only during *ex vivo* expansion for 5 days, and removed before T cell transduction and injection into the mice. We have now revised Figure 5, Figure 6, as well as the main text to make it clear that the main aim of the study was to identify small molecule inhibitors that can be used to improve T cell cytotoxicity without having to be co-injected with the T cell product.

Minor Concerns:

In the figure legend of Figure 3C, CCL28 should be CCL23.

Response:

Thank you, this has been corrected.

Should CTLA4 be a T cell costimulatory molecule or immune checkpoint effector?

Response:

CTLA4 blockade has been effective at reinvigorating immune response in some clinical settings, and CTLA4 is generally recognized as an inhibitory receptor to T cell effector function. However, inhibitory receptors such as CTLA4, PD1, TIM3, and LAG3 are also co-expressed during T cell activation and differentiation (as reviewed in PMID: 26167163 and PMID: 26205583). It is increasingly recognized that individual modest gains in inhibitory receptors are not sufficient to conclude T cell exhaustion. Instead, exhaustion might be better characterized by constitutively high expression of multiple inhibitory receptors, critical loss of cytokine expression, proliferation and cytotoxicity.

REVIEWERS' COMMENTS

Reviewer #1 (Remarks to the Author):

I've been reading the revised manuscript and the response provided to the reviewers' concerns. I consider that the manuscript has been adequately improved and now merits acceptance in Nature Comm.

Reviewer #2 (Remarks to the Author):

The reviewers extensively revised their manuscript and performed additional experiments. I have no further major comments. In general and throughout the manuscript, it would be preferable if the authors could show all data points as opposed to error bars.

Reviewer #3 (Remarks to the Author):

The authors have adequately addressed my concerns.